# Sharp description of local minima in the loss landscape of high-dimensional two-layer ReLU neural networks

Jie Huang [1]    Bruno Loureiro [2]    Stefano Sarao Mannelli [3 4]

## Abstract

We study the population loss landscape of two-layer ReLU networks of the form $\sum_{k=1}^{K} \mathrm{ReLU}(w_k^\top x)$ in a realisable teacher–student setting with Gaussian covariates. We show that local minima admit an exact low-dimensional representation in terms of *summary statistics*, yielding a sharp and interpretable characterisation of the landscape. We further establish a direct link with one-pass SGD: local minima correspond to attractive fixed points of the dynamics in summary statistics space. This perspective reveals a hierarchical organisation of minima into discrete families and shows how overparameterisation changes their stability and reachability under gradient-based dynamics. In this overparameterised regime, global minima become increasingly accessible, attracting the dynamics and reducing convergence to spurious solutions. Overall, our results reveal intrinsic limitations of common simplifying assumptions, which may miss essential features of the loss landscape even in minimal neural network models.

## 1. Introduction

Modern neural networks are trained by first-order algorithms on loss functions that are highly non-convex, structured, and shaped by both the architecture and the data distribution. Nevertheless, despite the inherent difficulty posed by non-convexity, day-a-day practice suggests that this is not a hindrance in the successful training of neural networks. Explaining this discrepancy requires a principled under-standing of the geometry of these landscapes: the structure of their critical points, the nature of their basins of attraction, and the global organisation of minima and saddles. A principled understanding of these landscapes and how descent-based algorithms navigate them is therefore a central challenge for developing a mathematical theory of deep learning.

Recent years have seen significant progress in this direction, most notably through results showing that the optimisation landscape of ReLU-based infinitely wide neural networks becomes benign in the mean-field limit, where global convergence guarantees can be established (Chizat & Bach, 2018; Mei et al., 2018; Rotskoff & Vanden-Eijnden, 2022; Sirignano & Spiliopoulos, 2020). However, such asymptotic regimes remain largely non-quantitative and do not capture the finite-width mechanisms by which landscape trivialisation emerges as overparametrisation is increased (Bach & Chizat, 2021). In particular, they offer limited guidance on how wide a network must be for these benign properties to hold, how landscape geometry changes as a function of model size, and which structural features of the data and target model control the onset of a benign optimisation regime.

In this work, we address these questions by studying the population risk landscape of teacher–student two-layer neural networks with ReLU activation under Gaussian inputs. In this setting, Safran & Shamir (2018) established the existence of spurious local minima and provided compelling evidence that increasing the network width leads to a simplification of the landscape ruling out the hypothesis of a uniformly benign finite-width landscape with a computer-assisted construction. In a follow-up work, Safran et al. (2021) formally proved that mild overparameterisation fundamentally alters the local Hessian, converting non-global minima into saddle points. Yet, empirical evidence clearly shows that the overparameterised landscape is not entirely benign, as higher-order local minima can form, leaving the geometry and dynamical role of these remaining traps largely unexplained by local analyses.

By contrast, in this work we develop a refined description of the population landscape by leveraging a summary-statistics representation of the population loss landscape. This approach allows us to explicitly characterise the geometry of minima and saddle points, to analyse their stability, and to

[1] Physics, Chalmers University of Technology and University of Gothenburg [2] Ecole Normale Superieure, PSL & CNRS [3] Data Science and AI, Computer Science and Engineering, Chalmers University of Technology and University of Gothenburg [4] School of Computer Science and Applied Mathematics, University of the Witwatersrand. Correspondence to: Stefano Sarao Mannelli <s.saraomannelli@chalmers.se>.

*Proceedings of the 43rd International Conference on Machine Learning*, Seoul, South Korea. PMLR 306, 2026. Copyright 2026 by the author(s).

track how the landscape deforms as a function of overparameterisation. In doing so, we provide a detailed account of how landscape trivialisation emerges, complementing existing existence results with a geometric and quantitative understanding of the underlying optimisation problem.

More precisely, our **main contributions** are:

- We give an exact characterisation of the population loss landscape of two-layer ReLU networks via a reduced set of self-consistent equations for low-dimensional *summary statistics*, see Result 1. This reduced formulation enables an efficient analytical description of the minima and clarifies their relationships as fixed points of gradient-flow on the population loss.

- Leveraging this description, we show that, in the well-specified setting where the teacher and student have equal width, the population loss of two-layer ReLU networks exhibits a hierarchy of local-minimum families at distinct loss levels. These minima are stable attractors for gradient-based algorithms and therefore constitute a significant obstacle to optimisation; see Fig.1a for an illustration.

- We demonstrate that overparameterisation alters the fixed-point structure and the resulting dynamics: low-order spurious minima are destabilised, higher-order families can persist, and the probability of reaching global minima increases sharply in the regimes we simulate.

- As shown in Fig.1b, these geometric properties directly govern the dynamics, which converge to quantised loss values characterised by the discrete hierarchy. Furthermore, the transition from the well-specified to the over-parameterised regime is reflected in the training statistics, where increasing the number of hidden units leads to a significantly larger fraction of trajectories reaching the global minimum.

Taken together, our results deliver a geometric characterisation of the population loss landscape of two-layer ReLU networks and its consequences for optimisation. We show how overparametrisation controls the emergence, stability, and suppression of spurious minima, thereby providing a principled fixed-point and dynamical explanation for the optimisation benefits of overparameterisation.

**Further related works**

**High-dimensional Dynamics and Mean-Field Theory.** The study of learning dynamics in two-layer neural networks using statistical physics techniques was pioneered by Biehl & Schwarze (1995); Saad & Solla (1995), who introduced the order-parameter description for soft committee machines. Recent works have rigorously established

the validity of these mean-field descriptions in the high-dimensional limit (Goldt et al., 2019; 2020). This framework has been extended to classify dynamical regimes based on the scaling of the learning rate and batch size (Ben Arous et al., 2022; Veiga et al., 2022; Arnaboldi et al., 2024), and to unify the description of single-pass, multi-pass SGD, and large-width (Arnaboldi et al., 2023b). While these works focus on the evolution of the risk, our work leverages this formalism to characterise the *fixed points* of these dynamics, specifically analysing the hierarchical structure of minima that emerges from the non-convex landscape.

**Optimisation Landscape of Two-Layer Networks.** A central question in deep learning theory is identifying conditions under which the loss landscape is benign. In the infinite-width limit, mean-field analyses (Mei et al., 2018; Chizat & Bach, 2018; Sirignano & Spiliopoulos, 2020; Rotskoff & Vanden-Eijnden, 2022) have shown that the loss landscape becomes convex-like, guaranteeing global convergence. However, this asymptotic result does not preclude the existence of spurious local minima in finite-width networks, as demonstrated empirically and theoretically by Safran & Shamir (2018); Safran et al. (2021) and Venturi et al. (2018). Our results bridge this gap by providing a quantitative description of these finite-width spurious minima and explicating the geometric mechanism by which overparameterisation simplifies the landscape, a phenomenon qualitatively discussed in Draxler et al. (2018); Simsek et al. (2021) and rigorously analysed for phase retrieval in Sarao Mannelli et al. (2020b); Davis et al. (2020). While Safran et al. (2021) relied on local Hessian evaluations to show that overparameterisation introduces negative eigenvalues at spurious minima, our summary-statistics approach provides a complementary, global view of the fixed-point families, their stability, and their dynamical reachability.

**Symmetry and Saddle Points Dynamics.** The permutation symmetry of hidden units plays a crucial role in shaping the loss landscape. Fukumizu & Amari (2000) originally identified that hierarchical plateaus in learning curves arise from symmetry-breaking transitions. This hierarchical learning manifests as a "staircase" profile, where the dynamics are governed by a separation of timescales between the learning of distinct features (Abbe et al., 2022; Jain et al., 2024; Berthier et al., 2025; Montanari & Urbani, 2025). In the specific context of ReLU networks, Arjevani & Field (2019) used group theory to show that spurious minima exhibit a "principle of least symmetry breaking," resulting in weight matrices with highly symmetric, block-like transitivity partitions. However, their discrete algebraic approach does not easily translate to generalisation dynamics. Our framework naturally recovers these symmetric blocks via the macroscopic order parameters (Eq. 7), linking them directly to the test error and learning trajectories. Related ge-

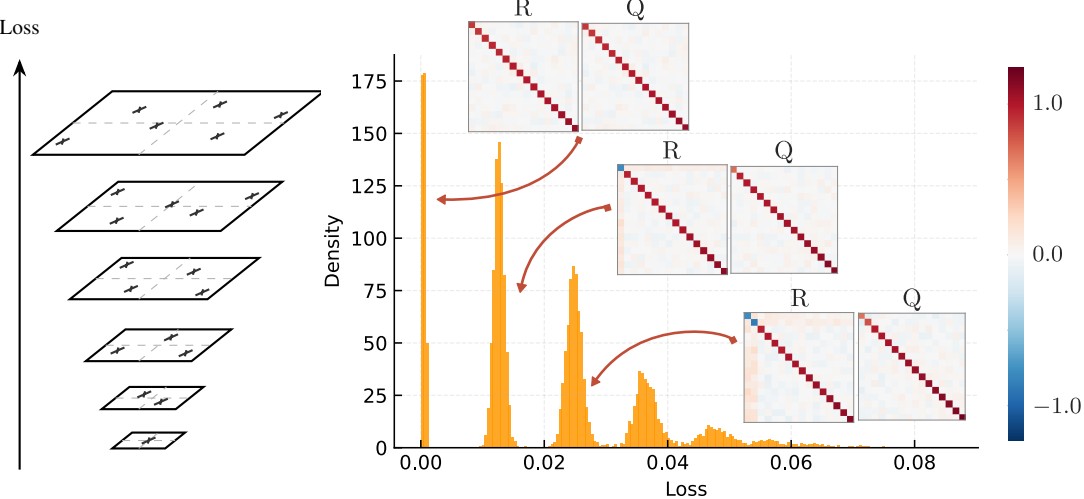

*(a) Local-minima families.*      *(b) Minima density and associated value of the order parameters.*

*Figure 1.* **Geometry and statistics of local minima. (a)** Schematic hierarchy of local-minimum families in the well-specified regime. Each plane represents a loss level and the crosses mark representative minima. **(b)** Validation of the theoretical predictions. The histogram shows the distribution of population risk reached by gradient flow ($10^4$ runs) in the high-dimensional standard normal teacher initialisations. Vertical dashed lines correspond to the analytical loss levels of the families derived in Result 2. Insets show heatmaps of the student-student ($Q$) and student-teacher ($R$) overlaps for a representative solution within the dominant family.

ometric analyses (Draxler et al., 2018; Garipov et al., 2018; Brea et al., 2019; Simsek et al., 2021) have highlighted that global landscape organisation can be substantially richer than what local curvature alone suggests. Similar questions have been explored in the context of the binary perceptron (Baldassi et al., 2019; 2021; Annesi et al., 2023; Barbier et al., 2024; Barbier, 2025). In the present work, we focus on the fixed-point structure and the dynamical role of saddle-like instabilities in ReLU teacher-student networks. While these saddle points do not affect the long-time asymptotic performance, they play a pivotal role in the transient, often slowing down optimisation. This dynamical role of saddles has been investigated in polynomial networks (Zhang et al., 2025) and classical statistical-physics models (Cugliandolo & Kurchan, 1993; 1994).

**Dynamics of Specialised and Simplified Architectures.**
Beyond the standard teacher-student setup, similar dynamical slowdowns and landscape complexities have been analysed in generalised linear models and multi-index models (Arous et al., 2021; Bietti et al., 2025; Şimşek et al., 2024), where the empirical risk can exhibit intricate spurious valleys (Maillard et al., 2020; Asgari et al., 2025). Furthermore, the mechanism of saddle-to-saddle dynamics has been largely explored in deep linear networks (Saxe et al., 2014; 2019). Recent results suggest that ReLU networks can exhibit similar behaviour: after an initial alignment phase (Boursier et al., 2022), the dynamics may follow a linear-network-like evolution, relating non-convex training to the deep linear network formalism (Saxe et al., 2022; Jarvis

et al., 2025). Finally, in models like phase retrieval (Chen et al., 2019; Mannelli et al., 2019; Arnaboldi et al., 2023a) and tensor PCA (Sarao Mannelli et al., 2019; 2020a), the structure of the Hessian (Bonnaire et al., 2024; 2025), the initialisation scheme (Jarvis et al., 2025; Annesi et al., 2025), and the temperature of the dynamics (Baity-Jesi et al., 2018) are known to dictate the escape from spurious traps. These traps are often organised in complex topologies, such as the "canyon" structures observed in high-dimensional chaos (Fournier et al., 2025) or the triplets of minima found in spiked tensor models (Ros et al., 2019; Pacco et al., 2025). Our work complements these findings by focusing on how these mechanisms manifest specifically in the ReLU nonlinearity via the compensation mechanism.

## 2. Problem Formulation and Summary Statistics Description

We consider the teacher-student framework, a convenient way of generating structured but tractable supervised learning tasks, where the target function $y = f_\star(x)$ — referred to as the *teacher* — is chosen from the same hypothesis class as the statistical model of interest — referred to as the *student*.

We focus on the class of two-layer neural networks with uniform readout layer and finite width. More precisely, each training example $(x^\mu, y^\mu)$ consists of an input vector $x^\mu \in \mathbb{R}^d$ and an output scalar $y^\mu$, where the components of $x^\mu$ are drawn i.i.d. from the standard normal distribution $\mathcal{N}(0, 1)$.

The target output is generated by a fixed *teacher* network with $M$ hidden units, $y^\mu = \phi(x^\mu, W^*)$, where $W^* \in \mathbb{R}^{M \times d}$ comprises the teacher weights $w_m^*$ (for $m = 1, \ldots, M$). Similarly, the *student* network approximates the target using $K$ hidden units and a parameter matrix $W \in \mathbb{R}^{K \times d}$ with rows corresponding to the weight vectors $w_k$. The teacher and student network functions are respectively defined as:

$$\phi(x, W^*) = \sum_{m=1}^{M} \text{ReLU}\left(\frac{w_m^{* \top} x}{\sqrt{d}}\right), \qquad (1)$$

$$\phi(x, W) = \sum_{k=1}^{K} \text{ReLU}\left(\frac{w_k^{\top} x}{\sqrt{d}}\right). \qquad (2)$$

We formally define the parameterisation regimes based on the relative width of student and teacher:

**Definition** (Parameterisation Regimes). *Let $M$ and $K$ denote the number of hidden units in the teacher and student networks, respectively.*

1. ***Well-specified:*** *The regime where $K = M$. In this case, the function space representable by the student coincides exactly with that of the teacher.*

2. ***Overparameterised:*** *The regime where $K > M$. In this case, the function space representable by the student includes and exceeds that of the teacher.*

Note that this definition can be different from the common usage of overparametrisation by practitioners, which typically compares the number of parameters in the model class to the amount of available data. Indeed, since the complexity of the target function is well-defined in our teacher-student task, it is natural to define overparametrisation with respect to this complexity.

The objective of the student network is to approximate the teacher function by minimising the discrepancy between their outputs. We define the *population risk* (or generalisation error) as the expected squared difference with respect to the input distribution:

$$\mathcal{L}(W; W^*) = \frac{1}{2}\mathbb{E}_x \left[(\phi(x, W) - \phi(x, W^*))^2\right]. \qquad (3)$$

The student learns by adjusting its parameters $W$ to minimise $\mathcal{L}(W; W^*)$ via gradient flow:

$$\dot{w}_k = -\eta \mathbb{E}_x \left[\mathcal{G}_k\right],$$
$$\mathcal{G}_k = (\phi(x, W) - \phi(x, W^*)) H\left(\frac{w_k^{\top} x}{\sqrt{d}}\right) \frac{x}{\sqrt{d}}, \qquad (4)$$

where $H(\cdot)$ denotes the Heaviside step function.

To analyse the system's dynamics, we introduce the relevant order parameters (or sufficient statistics) in the form of weight overlaps:

$$Q_{ij} = \frac{1}{d}w_i^{\top} w_j, \; R_{im} = \frac{1}{d}w_i^{\top} w_m^*, \; T_{mn} = \frac{1}{d}w_m^{* \top} w_n^*, \qquad (5)$$

with $Q \in \mathbb{R}^{K \times K}$, $R \in \mathbb{R}^{K \times M}$, and $T \in \mathbb{R}^{M \times M}$. For analytical clarity, we assume orthonormal teacher weights, such that $T = I_M$ (the identity matrix). This assumption simplifies the discussion without loss of generality, with the same phenomenology as long as $T$ is full-rank. Our analysis extends to arbitrary teacher configurations and supporting results are provided in Appendix C.

**Result 1** (Necessary Conditions for Minima). *The stationary points of the population risk correspond to the zeros of the gradient flow. These satisfy a system of coupled nonlinear equations governing the order parameters $Q$ and $R$:*

$$\mathcal{F}_R(Q, R) = 0, \qquad \mathcal{F}_Q(Q, R) = 0. \qquad (6)$$

*The explicit, analytical forms of the functionals $\mathcal{F}_R$ and $\mathcal{F}_Q$ are detailed in Appendix B.*

Notably, these equations are exact and independent of the input dimension $d$. We discuss the broader generality of these results in Sec. 3.2.

**Result 2** (Classification of Local Minima). *The local minima are not unique; they are organised into distinct families characterised by a structural parameter $k_1 \in [0, M]$, representing the specific count of student units that become anti-aligned with the teacher vectors.*
*By imposing a block-symmetric ansatz based on $k_1$, the general conditions in Result 1 reduce to a tractable set of equations (derived in Appendix B). Where real solutions exist, this allows us to analytically determine the order parameters $Q(k_1)$ and $R(k_1)$ and directly compute macroscopic observables like the exact generalisation error.*

This structural parameter $k_1$ directly maps to the transitivity partitions of the isotropy subgroups identified by Arjevani & Field (2019). Our macroscopic reduction confirms that their discrete algebraic classes of spurious minima dictate the quantised levels of the population risk.

Fig. 1b illustrates the consequences of these findings and serves as a visual guide to the results presented in Sec. 3.

*Landscape Structure (Fig. 1a):* The local minima are organised into families with distinct loss values and symmetry-breaking patterns. Fig. 1a gives a schematic representation of these families in the well-specified regime; the quantitative structure is described through the order parameters and their fixed-point equations.

*Dynamical Selection (Fig. 1b):* The existence of these families strongly affects the dynamics as discussed in Sec. 3.2.

The figure shows the density of solutions found by gradient flow across $10^4$ initialisations. This highlights the sharp concentration of solutions around specific loss levels and validates our theoretical predictions (vertical dashed lines). The insets display the structure of the order parameters $Q$ and $R$ for typical solutions, highlighting the symmetry-breaking patterns characteristic of each family.

## 3. Landscape geometry and consequences for optimisation

We leverage our theoretical framework to dissect the optimisation landscape. To do so, we rely on the numerical integration of the derived ODEs for the order parameters $Q$ and $R$. This allows us to simulate the mean-field trajectories, which we use to complement, explain, and validate the behaviour of the simulations.

### 3.1. Landscape Characterisation

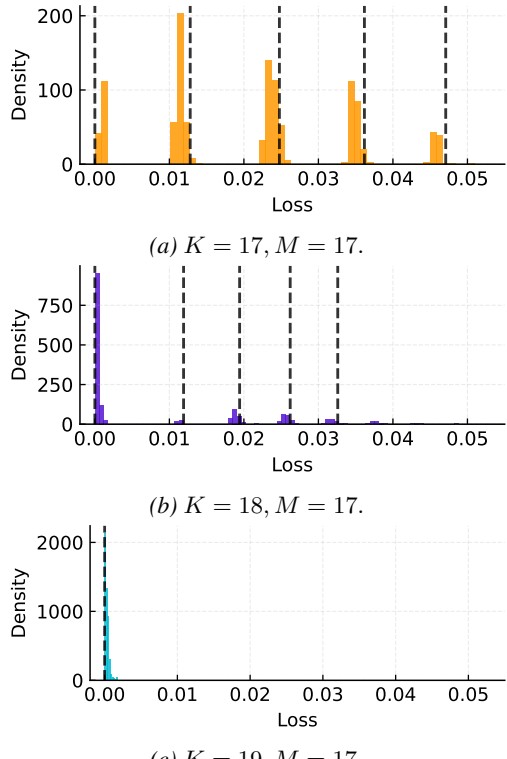

*(a) $K = 17, M = 17$.*

*(b) $K = 18, M = 17$.*

*(c) $K = 19, M = 17$.*

*Figure 2.* **Loss families and theoretical value.** Histogram of final loss values obtained from ODE dynamics at $M = 17$ for $K = M$ (top), $K = M + 1$ (middle), and $K = M + 2$ (bottom) starting from $10^4$ initialisations in orthonormal teacher configuration. Dashed vertical lines indicate the corresponding theoretical loss values obtained from the Result 2 For $K = M$ and $K = M + 1$, the theoretical values capture the locations of the local minima families and the minor discrepancies may be attributed to numerical effects. In contrast, for $K = M + 2$, local minima are strongly suppressed in $M = 17$ and the loss distribution concentrates near zero.

**Spurious minima exist and are reachable.** We start our investigation of the population loss landscape by the well-specified case where both teacher and student have the same width ($K = M$) — the minimal case for which the model has the capacity required to approximate the target. The evolution of the weights follows the gradient flow described in Eq. (4). By adopting the point of view of the order parameters, we can write a closed set of ordinary differential equations (ODEs) governing the time evolution of $Q(t)$ and $R(t)$, see Appendix A. In Fig.1b and Fig.2, we simulate these dynamics numerically starting from $10^4$ independent random initialisations drawn from the standard normal distribution. Consistent with the empirical results by Safran & Shamir (2018), we observe that in the well-specified setting the gradient flow frequently becomes trapped in suboptimal stationary points with non-vanishing generalisation error. As shown in the Figures, the distribution of final loss values exhibits a quantised structure: steady states do not populate a continuum but rather concentrate tightly around discrete loss levels, indicating that the landscape is dominated by distinct families of spurious minima.

To understand the geometric origin of these families, we examine the spectral properties of the order parameters. Due to the permutation symmetry of the hidden units, the network and loss are invariant under reordering of the student neurons. As exemplified in the insets of Fig.1b and discussed in detail in Appendix B.1, we observe that the discrete loss levels correspond to specific symmetry-breaking configurations where a subset of $k_1$ student neurons becomes *anti-aligned* with the teacher vectors (i.e., $R_{im} < 0$), a structure made explicit in Fig. 7. This anti-alignment induces a large local error, which triggers a *compensation mechanism* in the remaining aligned neurons. Specifically, the aligned units adjust their orientation to offset the negative contribution of the anti-aligned units, effectively zeroing out the gradient. This cooperative effect stabilises the system at a local minimum, preventing the anti-aligned units from flipping back to the correct orientation under the dynamics. As the number of anti-aligned units increases, the required compensation grows until the configuration becomes unstable; thus, these stable minima can be naturally ordered by the integer count of anti-aligned neurons, defining an hierarchy between them.

This empirical regularity can be analytically formalised by imposing a block-symmetric ansatz on the order parameters $(R, Q)$. We partition the student hidden units into two groups: a set $I_1$ of size $k_1$ (anti-aligned) and a set $I_2$ of size $k_2$ (aligned), such that $k_1 + k_2 = K$. We visualise this ansatz in Eq. (7), where we adopt the notation $\mathbf{B}(x, y)$ to represent a block matrix with diagonal elements $x$ and

off-diagonal elements $y$ (i.e., $xI + y(J - I)$).

$$R = \left( \begin{array}{c|c} \mathbf{B}(r_1^{\text{diag}}, r_1^{\text{off}}) & r_{12}^{\text{cross}} \\ \hline r_{21}^{\text{cross}} & \mathbf{B}(r_2^{\text{diag}}, r_2^{\text{off}}) \end{array} \right),$$

$$Q = \left( \begin{array}{c|c} \mathbf{B}(q_1^{\text{diag}}, q_1^{\text{off}}) & q_{12}^{\text{cross}} \\ \hline q_{12}^{\text{cross}} & \mathbf{B}(q_2^{\text{diag}}, q_2^{\text{off}}) \end{array} \right). \quad (7)$$

Here, the blue blocks correspond to the $k_1$ anti-aligned units, the green blocks to the $k_2$ aligned units, and the grey blocks capture the interaction between the two populations. This specific block-symmetric structure of the order parameters serves as the macroscopic, statistical-mechanics equivalent of the discrete isotropy subgroups identified by Arjevani & Field (2019), confirming that the "principle of least symmetry breaking" dictates the clustering of the network's weights.

In Appendix B, we report additional supporting validation of the ansatz.

**The Role of Over-parameterisation.** Having characterised the spurious minima in the well-specified case, we turn our attention to how overparametrisation, i.e. increasing the student width to $K > M$, alters the landscape and facilitates convergence.

**Global Convergence and Stability.** To quantify the benefit of overparameterisation, we track the optimisation trajectories of $10^3$ random initialisations under gradient flow across three regimes: well-specified ($K = M$) and mildly overparameterised ($K = M + 1$ and $K = M + 2$). As illustrated in Fig.2a, in the well-specified setting a significant fraction of trajectories remain trapped in the high-loss families identified in Sec.3.1. In contrast, the addition of even a single extra neuron ($K = M + 1$), Fig.2b-c, dramatically expands the basin of attraction of the global minimum, with nearly all trajectories converging to zero loss. These empirical results indicate that overparameterisation effectively destabilises the spurious minima that plague the $K = M$ landscape.

We interpret this improvement through the stability of fixed points. The addition of extra neurons gives perturbations access to directions that are absent in the well-specified model. In particular, the zero-padding or neuron-splitting construction analysed by Fukumizu & Amari (2000); Safran et al. (2021) predicts that non-global minima of the $K = M$ landscape can acquire negative-curvature directions once embedded into a wider network. The perturbative experiment below probes this mechanism directly in the summary-statistics dynamics.

When a fixed point is embedded into the wider landscape by adding an inactive neuron, it is crucial to determine whether the embedded point remains an attractor or becomes unstable. Standard Hessian analysis is ill-suited here due to the

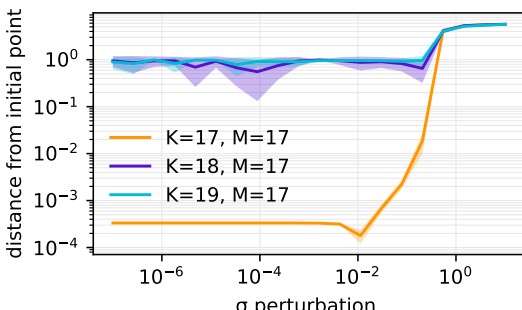

*Figure 3.* **Perturbative analysis of fixed points.** Distance from the initial configuration obtained by adding a Gaussian perturbation of standard deviation $\sigma$ in weight space. The plot consider the average distance after the network has been allowed to relax with gradient descent for $1,000$ steps and learning rate $0.01$. The mean (solid line) and standard deviation (shaded area) are evaluated by collecting 20 independent perturbations. The different line represent different level of over-parameterisation. The exact-parametirisation case appears to be less that $0.001$ from the original configuration, a distance that is compatible with the limited epochs allowed by gradient descent, and it is therefore considered stable from small enough $\sigma$. On contrary, the overparameteritdsed case show instability even at small $\sigma$.

non-differentiability of the ReLU activation. Instead, we rely on a perturbative dynamical analysis. We initialise the system at a fixed point and apply a Gaussian perturbation to the weights: $W \to W + \xi$, where $\xi \sim \mathcal{N}(0, \sigma^2 I)$. Note that we perturb the weights rather than the order parameters directly, as arbitrary perturbations to $Q$ and $R$ may violate the geometric constraints (details in Appendix D). We then allow the system to relax via gradient flow. The results, reported in Fig.3, demonstrate that, while in the well-specified case spurious fixed points are stable to perturbations, i.e. the dynamics consistently return to the attractor. Conversely, in the over-parameterised case ($K \geq M + 1$), these same fixed points become unstable. The perturbation excites the extra neuron, allowing the system to escape the saddle point.

**Fixed Points in the Overparameterised Regime.** We now extend our fixed-point analysis to the overparameterised setting ($K = M + 1$). We generalise the ansatz (Result 2) to accommodate the additional degrees of freedom, identifying stationary points where the extra neurons carry non-trivial weight (details in Appendix B).

This revealed that in the overparameterised landscape, the lowest-order spurious minima are destabilised. Specifically, for the family characterised by a single anti-aligned neuron ($k_1 = 1$), we are not able to find stable solutions to the fixed-point equations. This theoretical prediction is corroborated by numerical integration of the ODEs reported in Table 1: across all initialisations, no trajectory converged to a state exhibiting a single anti-aligned direction. However, higher-order spurious minima (with $k_1 \geq 2$) persist. These fami-

lies, which correspond to more complex symmetry-breaking configurations, remain stable even in the presence of the additional neuron. This paints a more complex picture with respect to Safran et al. (2021). Indeed, while their results established that overparameterisation can destabilise local minima via simple neuron splitting (provided the weight norms are bounded—a condition naturally satisfied by our macroscopic order parameters, where $\sum_i \|w_i\| \approx M - k_1$), our global analysis reveals a much more intricate landscape. We find that while simple, uncoupled defects ($k_1 = 1$) are indeed annihilated by the addition of a single neuron, higher-order spurious minima ($k_1 \geq 2$) actively persist. Crucially, these surviving local minima are not merely artifacts of adding a zero-weight neuron to an exact-parameterisation solution; they are complex, coupled structures that emerge uniquely in the overparameterised space. Because they cannot be derived through simple zero-padding or local Hessian perturbations, identifying these surviving traps fundamentally requires solving the exact macroscopic fixed-point equations.

| $Minima$ $Order$ | $K = 17$ $M = 17$ | $K = 18$ $M = 17$ | $K = 19$ $M = 17$ |
|---|---|---|---|
| $k_1 = 0$ | 13.09% | 59.29% | 99.63% |
| $k_1 = 1$ | 27.52% | 0.00% | 0.00% |
| $k_1 = 2$ | 29.05% | 2.10% | 0.05% |
| $k_1 = 3$ | 18.94% | 10.83% | 0.31% |
| $k_1 = 4$ | 7.55% | 8.99% | 0% |

*Table 1.* Percentage of $10^4$ random initialisations that converge to minima with $k_1$ anti-aligned units in different $(K, M)$; $k_1 = 0$ corresponds to the global minimum.

Thus, overparameterisation does not simply remove all spurious structure. Instead, it changes which fixed-point families are stable and how frequently gradient flow reaches them. This reveals a richer overparameterised landscape than what is captured by local Hessian arguments alone.

### 3.2. Dynamical Characterisation

As discussed in the problem formulation Sec.2, the evolution of the weights is governed by a closed set of ODEs for the order parameters, $\dot{Q} = \mathcal{F}_Q(Q, R)$ and $\dot{R} = \mathcal{F}_R(Q, R)$. While the full explicit forms are deferred to Appendix A, the dynamics can be written in terms of Gaussian expectation terms whose form depends on the activation function. For ReLU, these admit closed-form expressions, given in Appendix A.4; corresponding derivations for Leaky ReLU and sigmoidal (erf) activations are provided in Appendix E.

In Fig.4, we numerically integrate the gradient flow ODEs to validate our theoretical landscape analysis. The simulations confirm that the minima identified with our ansatz are important attractors of the problem. Indeed, the loss trajectories do not settle into a continuum, but rather concentrate into

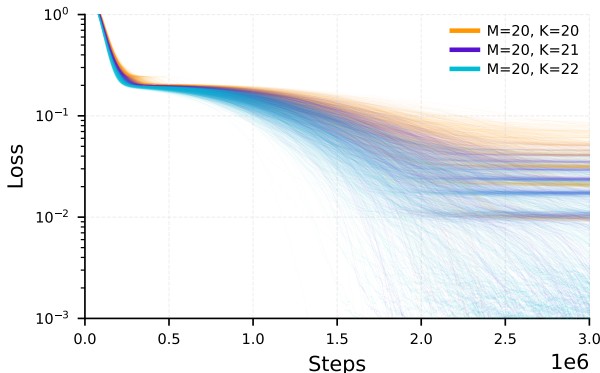

*Figure 4.* **Training dynamics and loss quantisation across parameterisation regimes.** Evolution of population risk under gradient flow for $M = 20$ teacher units (1,000 random initialisations per condition, learning rate $\eta = 0.1$ , orthonormal teacher configuration). All regimes exhibit entrapment in discrete high-loss plateaux. However, while the well-specified case (K=20, orange) is dominated by these suboptimal attractors, mild over-parameterisation ($K = 21$, purple; $K = 22$, blue) progressively destabilises intermediate plateaus, allowing a larger fraction of trajectories to escape towards the global minimum. The observed loss bands correspond to the theoretical loss levels of the spurious families predicted by Result 2.

discrete, quantised bands. These plateaus align precisely with the loss levels of the spurious families predicted by Result 2. Furthermore, comparing the regimes ($K = M$ vs. $K > M$) illustrates the dynamical advantage of overparameterisation, which destabilises some of the intermediate bands and facilitates the cascade toward the global minimum.

**Alternative Optimisation Schemes.** While our primary analysis focuses on gradient flow, our framework is flexible and can be readily adapted to alternative descent-based algorithms often studied in theoretical literature or employed in practice. We derive the corresponding modified ODEs for the following settings in Appendix A and Appendix F:

- *Normalized GD (nGD).* Optimisation is performed on the hypersphere with a fixed norm constraint, $\|w_k\|_2 = \sqrt{d}$. The dynamics project the gradient onto the tangent space of $\mathbb{S}^{d-1}$:

$$\dot{w}_k = -\eta \mathbb{E}[\mathcal{G}_k] + \frac{\eta}{d} \langle \mathbb{E}[\mathcal{G}_k], w_k \rangle w_k. \quad (8)$$

- *Orthonormalized GD (onGD).* The weights are constrained to the Stiefel manifold, ensuring the student weight matrix remains orthonormal ($WW^\top = d\, I_K$). The update rule projects the gradient via $\mathcal{P}_W(Z) = Z - \frac{1}{d} ZW^\top W$:

$$\dot{W} = -\eta \mathcal{P}_W(\mathbb{E}[\mathcal{G}]). \quad (9)$$

- *Two-Layer GD (2L-GD).* Both student weights $w_k$ and readout coefficients $v_k$ are trained simultaneously. The continuous-time dynamics are coupled:

$$\dot{w}_k = -\eta \mathbb{E}[\mathcal{G}_k^w] \quad \text{and} \quad \dot{v}_k = -\eta \mathbb{E}[\mathcal{G}_k^v]. \qquad (10)$$

Consequently, the flow equations must be augmented to include the evolution of the second layer, i.e., $\dot{Q} = \mathcal{F}_Q(Q, R, v)$.

- *Online Stochastic Gradient Descent (SGD)*. Optimisation is performed on the empirical loss using new samples at each step: $\Delta w_k = -\eta \mathcal{G}_k$. While formally a discrete process, in the limit of high-dimensional inputs ($d \to \infty$), the dynamics can be described by a deterministic set of ODEs (see Appendix A.6).

A key question concerns the generality of the landscape properties derived under gradient flow.

**Result 3** (Landscape Equivalence for SGD). *If the learning rate scales as $\eta = o_d(1)$, the additional diffusive terms in the ODEs for SGD vanish (see Eqs. 29-31 Appendix A.6). In this regime, the trajectory of SGD converges to that of the gradient flow. Consequently, the fixed-point and dynamical analysis presented in previous sections applies directly to SGD in the appropriate scaling limit.*

**Empirical Comparison and Convergence.** We compare the convergence properties of these optimisers in Table 2, aggregating statistics from $10^4$ initialisations. The results highlight a sharp distinction between unconstrained (GD, 2L-GD) and constrained (nGD, onGD) dynamics.

| Optimiser | $K = 17$ $M = 17$ | $K = 18$ $M = 17$ | $K = 19$ $M = 17$ |
|---|---|---|---|
| GD | 13.25% | 64.18% | 77.50% |
| 2L-GD | 13.24% | 67.91% | 99.48% |
| nGD | 14.12% | 58.35% | N/C |
| onGD | N/C | N/C | N/C |

*Table 2.* **Convergence frequency across optimisation schemes.** The table reports the percentage of runs reaching the global minimum out of $10^4$ initialisations. Columns represent different parameterisation regimes (exact vs. over-parameterised). Consistent with Safran & Shamir (2018), over-parameterisation aids convergence in unconstrained dynamics. A more thorough comparison is reported in Appendix G. However, constrained dynamics (nGD, onGD) show severe slowdowns, resulting in non-convergence (N/C) within the allocated computational budget ($1.2 \times 10^7$ steps), a comprehensive analysis is detailed in Appendix G.1.

First, constrained dynamics exhibit significantly slower convergence timescales, often failing to reach steady states even after extended training (denoted as N/C in Table 2). To provide a complete picture of these failure modes, Appendix G.1 provides the full classification tables for nGD across all network widths, alongside the unquantised final loss distributions characteristic of onGD. Second, the nature of the landscape for onGD is fundamentally different. The compensation mechanism identified in Sec.3.1 relies on the ability of aligned neurons to adjust their magnitude to cancel the error from anti-aligned units. The orthonormality constraint forbids such magnitude adjustments. Therefore, the specific families of spurious minima discussed previously cannot exist in the onGD setting, rendering its landscape distinct from the standard ReLU network landscape.

Conversely, nGD recovers a minima structure analogous to GD. However, in the overparameterised regime ($K = M + 2$), we observe a high frequency of "mixed" states characterised by combinations of minima from different families. While these states are theoretically unstable, their gradients suggest proximity to saddle points Zhang et al. (2025), causing severe dynamical slowdowns. This accounts for the lower convergence rates compared to GD and 2L-GD.

## 4. Conclusions

In this work, we provided a sharp characterisation of the population loss landscape for high-dimensional two-layer ReLU networks, bringing together perspectives from dynamical approaches and landscape analysis and shedding new light on open problems left by Safran & Shamir (2018). Our analysis identifies a hierarchy of spurious-minimum families in the well-specified regime and shows that over-parameterisation changes their stability and reachability: low-order traps can disappear, higher-order traps can persist, and global minima become more frequently reached by gradient flow. Leveraging mean-field theory, we confirmed that these landscape features act as strong attractors for the gradient flow, governing the "quantised" loss distributions observed empirically. We further established the generality of these findings, extending the validity of this landscape description to SGD under appropriate scaling. By doing so, our macroscopic framework bridges two distinct perspectives in the literature: the group-theoretic symmetries of spurious minima (Arjevani & Field, 2019) and the local Hessian instabilities triggered by neuron splitting (Safran et al., 2021). We show that these static, finite-width geometric properties govern the global, high-dimensional learning trajectories. Key questions remain for future investigation. Future work includes quantifying the basins of attraction for different initialisations and developing a sharper global geometric description of the fixed-point families. Additionally, extending this analysis to a broader class of non-linearities and optimisation protocols offers a promising future direction. Overall, this work establishes a rigorous, interpretable framework for understanding how over-parameterisation mitigates the inherent non-convexity of neural network training.

## Acknowledgments

The authors would like to acknowledge discussions with Sebastian Goldt, Francesca Mignacco, Ashkan Panahi, Courtney Paquette, Berfin Simsek, and Ludovic Stephan concerning this problem. S.S.M. was supported by the Wal-

lenberg AI, Autonomous Systems, and Software Program (WASP). B.L. was supported by the French government, managed by the National Research Agency (ANR), under the France 2030 program with the project references "ANR-23-IACL-0008" (PR[AI]RIE-PSAI) and "ANR-25-CE23-5660" (MAPLE), as well as the Choose France - CNRS AI Rising Talents program. The computations were enabled by resources provided by the National Academic Infrastructure for Supercomputing in Sweden (NAISS) at Alvis (Chalmers Centre for Computational Science and Engineering, C3SE) and Tetralith, partially funded by the Swedish Research Council through grant agreement no. 2022-06725, under project NAISS 2024/22-1082.

## Accessibility

We are committed to ensuring the accessibility of this manuscript. All figures have been designed using a high-contrast, colourblind-friendly palette to guarantee readability for individuals with colour vision deficiencies.

## Software and Data

We provide a comprehensive description of the theoretical framework and numerical methodologies to ensure the full reproducibility of our results. All equations, derivation steps, and simulation hyperparameters are explicitly detailed in the main text and appendices. The complete source code for the mean-field dynamics and fixed-point solvers will be made publicly available upon publication.

## Impact Statement

This paper presents work whose goal is to advance the theory of Machine Learning, specifically regarding the optimisation landscape of neural networks. Given the theoretical nature of our contribution, we do not see immediate potential societal consequences.

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

# Appendix

This appendix provides detailed derivations, algorithmic descriptions, and additional experimental results supporting the Main Text. The content is organised as follows:

- **Appendix A: Dynamics.**
  Full derivation of the mean-field ODEs for Gradient Descent, the calculation of relevant Gaussian integrals ($I_2, I_3$), and the extension to Stochastic Gradient Descent.

- **Appendix B: Equations of Fixed Points.**
  Derivation of the implicit equations for stationary points, the block-symmetric ansatz formulation, and numerical verification of the solutions.

- **Appendix C: Robustness to Finite Input Dimensionality.**
  Analysis of the impact of finite input dimensions ($d$) on the loss distribution and the structural robustness of the macroscopic order parameters.

- **Appendix D: Stability of fixed points from higher order minima.**
  Perturbative dynamical analysis examining the stability of fixed points and the escape mechanisms from saddle points.

- **Appendix E: Alternative Activation Functions.**
  Analytical derivation of integrals and landscape analysis for Leaky ReLU and Sigmoidal (erf) activations.

- **Appendix F: Constraint Dynamics.**
  Derivations of the mean-field equations for Normalized GD (nGD) and Orthonormalised GD (onGD), along with extended empirical results (Appendix G.1) analysing their convergence properties and unquantised loss distributions.

- **Appendix G: Spurious minima in two-layer ReLU networks.**
  Tabulated statistics of global versus local minima across different parameterisation regimes for unconstrained gradient descent, validating agreement with prior empirical works.

- **Appendix H: Numerical and Computational Details.**
  Summary of the computational setup, including hardware, parallelisation strategy, and runtime requirements for the numerical experiments.

---

## A. Dynamics

As shown in Fig. 5, the mean-field ODEs provide an accurate description of the loss dynamics observed in simulations of finite-width neural networks trained by gradient descent. We now provide the derivation of these equations.

### A.1. Population Gradient

We provide additional details on the ODEs for our learning problems. The network is trained using Gradient Descent (GD), Normalized GD (nGD), or Orthonormalized GD (onGD) on the MSE loss function. In this section, we focus on the derivation for standard GD with ReLU activation functions. The derivations for nGD and onGD are detailed in Appendix F, and integrals for other activations (e.g., Leaky ReLU, erf) are provided in Appendix E.

Following the notation used by Goldt et al. (2019), we use indices $i, j, k, l = 1, \ldots, K$ for student units and $n, m = 1, \ldots, M$ for teacher units.

For the sake of completeness, we discussed the general two-layer network case, were both first layer ($W$) and second layer ($V$) are trained. Eventually, we will ignore the update of the second layer for the case where this is relevant. Additionally, we will consider a generic activation function $g$ and specialise on individual activation differences later on.

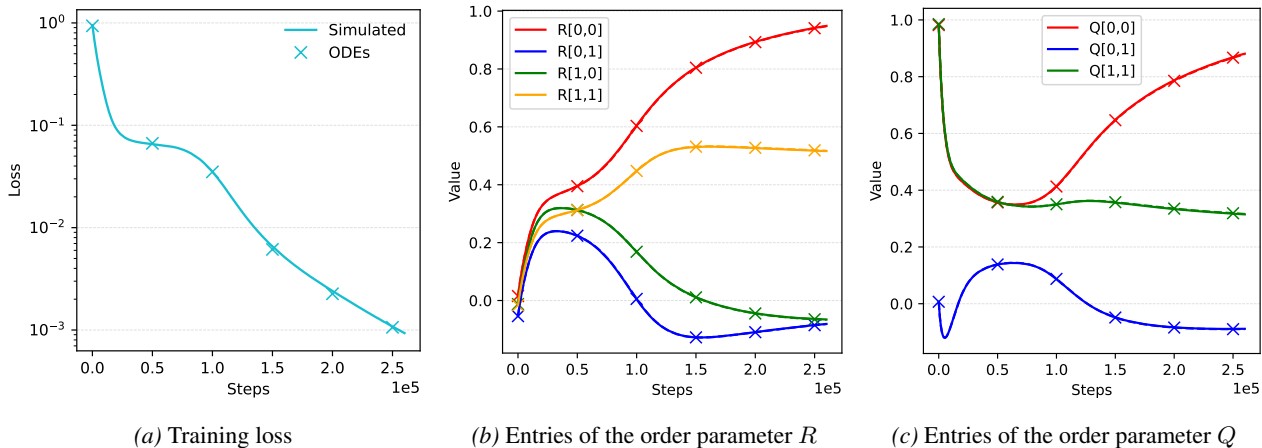

*(a)* Training loss          *(b)* Entries of the order parameter $R$          *(c)* Entries of the order parameter $Q$

*Figure 5.* **Comparison between mean-field ODE predictions and network simulations.** Panel (a) shows the training loss, while panels (b) and (c) display representative entries of the order parameters $R$ and $Q$ as functions of training steps. Solid lines denote simulation results and markers indicate ODE predictions.

We express the loss function between student and teacher networks as:

$$\mathcal{L} = \frac{1}{2}\left\langle \left[\sum_{k=1}^{K} v_k g(\lambda_k) - \sum_{m=1}^{M} v_m^* g(\rho_m)\right]^2 \right\rangle, \tag{11}$$

where $\lambda_k = \frac{w_k^\top x}{\sqrt{d}}$ and $\rho_m = \frac{w_m^{*\top} x}{\sqrt{d}}$ are the pre-activations. The continuous-time dynamics of the student weights $w_i$ under gradient descent are given by:

$$\frac{dw_i}{dt} = -\eta \nabla_{w_i} \mathcal{L} = -\eta\, v_i \left\langle \Delta\, g'(\lambda_i) \frac{x}{\sqrt{d}} \right\rangle, \tag{12}$$

$$\frac{dv_i}{dt} = -\eta \nabla_{v_i} \mathcal{L} = -\eta \left\langle \Delta\, g(\lambda_i) \right\rangle, \tag{13}$$

where $\Delta = \phi(x) - \phi^*(x) = \sum_j v_j g(\lambda_j) - \sum_m v_m^* g(\rho_m)$ is the error signal, and $\eta$ is the learning rate for weights.

## A.2. Derivation of Dynamics

The pre-activations $\lambda$ and $\rho$ are joint multivariate normal distributions with zero mean. The covariance matrix $\Sigma$ is determined by the order parameters already reported in Eqs. 5 of the Main Text:

$$Q_{ik} = \frac{w_i^\top w_k}{d} = \langle \lambda_i \lambda_k \rangle, \quad R_{in} = \frac{w_i^\top w_n^*}{d} = \langle \lambda_i \rho_n \rangle, \quad T_{nm} = \frac{w_n^{*\top} w_m^*}{d} = \langle \rho_n \rho_m \rangle.$$

To derive the closed-form ODEs, we apply the chain rule to the definition of the order parameters. For the student-student overlap $Q_{ik} = \frac{1}{d} w_i^\top w_k$:

$$\frac{dQ_{ik}}{dt} = \frac{1}{d}\left(\frac{dw_i}{dt}^\top w_k + w_i^\top \frac{dw_k}{dt}\right). \tag{14}$$

Substituting the weight dynamics from Eq. (12) into the first term, we obtain:

$$\frac{1}{d}\frac{dw_i}{dt}^\top w_k = -\frac{\eta}{d} v_i \left\langle \Delta \cdot g'(\lambda_i) \frac{x^\top w_k}{\sqrt{d}} \right\rangle = -\frac{\eta}{d} v_i \left\langle \Delta \cdot g'(\lambda_i)\lambda_k \right\rangle. \tag{15}$$

The remaining factor $1/d$ represents the intrinsic time scale of the mean-field dynamics. By absorbing this factor into the continuous time variable (i.e., defining the macroscopic time step $dt \sim \frac{1}{d}$), we obtain the $O(1)$ dynamical equations:

$$\frac{dQ_{ik}}{dt} = -\eta v_i \langle \Delta g'(\lambda_i)\lambda_k \rangle - \eta v_k \langle \Delta g'(\lambda_k)\lambda_i \rangle, \tag{16}$$

where $\eta$ denotes the effective learning rate [1]. The second term follows by symmetry ($i \leftrightarrow k$). Expanding the error signal $\Delta$, we arrive at terms of the form $\langle g'(\lambda_i)\lambda_k g(\lambda_j)\rangle$, which motivates the definition of the integral $I_3$. This formulation applies generally to two-layer networks with activation function $g(\cdot)$ and hidden layer width not larger than $\mathcal{O}(d)$. See (Arnaboldi et al., 2023b) for a detailed discussion.

Similarly, for the student-teacher overlap $R_{in}$:

$$\frac{dR_{in}}{dt} = \frac{1}{d}\frac{dw_i}{dt}^\top w_n^* = -\frac{\eta}{d}v_i\langle \Delta \cdot g'(\lambda_i)\rho_n\rangle \implies -\eta v_i\langle \Delta \cdot g'(\lambda_i)\rho_n\rangle. \tag{17}$$

Finally, for the second-layer weights $v_i$, the gradient descent dynamics are given by:

$$\frac{dv_i}{dt} = -\eta\langle \Delta \cdot g(\lambda_i)\rangle. \tag{18}$$

Substituting the explicit form of the error signal $\Delta = \sum_{j=1}^K v_j g(\lambda_j) - \sum_{m=1}^M v_m^* g(\rho_m)$, we obtain:

$$\frac{dv_i}{dt} = -\eta\left[\sum_{j=1}^K v_j\langle g(\lambda_j)g(\lambda_i)\rangle - \sum_{m=1}^M v_m^*\langle g(\rho_m)g(\lambda_i)\rangle\right]. \tag{19}$$

Here, the expectations are taken over the joint Gaussian distribution of the fields. We observe terms involving the product of two activation functions, $\langle g(\cdot)g(\cdot)\rangle$, which motivates the definition of the two-variable integral $I_2(x_1, x_2) \equiv \langle g(x_1)g(x_2)\rangle$.

### A.3. Full Equations

The closed-form ODEs training for two layers are:

$$\frac{dR_{in}}{dt} = \eta v_i\left[\sum_{m=1}^M v_m^* I_3(i, n, m) - \sum_{j=1}^K v_j I_3(i, n, j)\right], \tag{20}$$

$$\frac{dQ_{ik}}{dt} = \eta v_i\left[\sum_{m=1}^M v_m^* I_3(i, k, m) - \sum_{j=1}^K v_j I_3(i, k, j)\right] + \eta v_k\left[\sum_{m=1}^M v_m^* I_3(k, i, m) - \sum_{j=1}^K v_j I_3(k, i, j)\right], \tag{21}$$

$$\frac{dv_i}{dt} = \eta\left[\sum_{m=1}^M v_m^* I_2(i, m) - \sum_{j=1}^K v_j I_2(i, j)\right]. \tag{22}$$

where we slightly abuse notation in the arguments of $I_2, I_3$ to explicitly show which variables are involved. For example, $I_3(i, n, m)$ implies $C_{00} = Q_{ii}, C_{11} = T_{nn}, C_{22} = T_{mm}, C_{01} = R_{in}$, etc.

### A.4. ReLU Activation Function Case

For ReLU, the Gaussian integrals have closed-form expressions. We collect here the explicit formulas used in the dynamics; corresponding expressions for other activations are given in Appendix E.

Let $(x_0, x_1, x_2)$ be jointly Gaussian variables with zero mean and covariance matrix $\Sigma$, where $C_{ij} = \langle x_i x_j\rangle$.

**Two-Variable Integral ($I_2$).** Used for the dynamics of the second-layer weights $v$.

$$I_2(x_1, x_2) = \langle g(x_1)g(x_2)\rangle. \tag{23}$$

For ReLU, explicit forms are given by:

$$I_2(\Sigma) = \frac{\sqrt{C_{11}C_{22} - C_{12}^2}}{2\pi} + \frac{C_{12}}{2\pi}\arccos\left(-\frac{C_{12}}{\sqrt{C_{11}C_{22}}}\right). \tag{24}$$

---

[1]Alternatively, one can absorb the entire $\eta/d$ term into the differential time, showing that it is enough to have $\eta$ or $1/d$ to obtain the ODEs. More discussion on this can be found in Arnaboldi et al. (2023b).

**Three-Variable Integral ($I_3$).** Used for the dynamics of $Q$ and $R$. We define $I_3$ specifically as the expectation involving a derivative, a multiplier, and an activation input:

$$I_3(x_0, x_1, x_2) = \langle g'(x_0)\, x_1\, g(x_2) \rangle. \tag{25}$$

For ReLU, $g'(x) = H(x) = \mathbf{1}_{x>0}$ (Heaviside step function). The analytic solution is:

$$I_3(\Sigma) = \frac{1}{2\pi}\left[ C_{12} \arccos\left( -\frac{C_{02}}{\sqrt{C_{22}C_{00}}} \right) + \frac{C_{01}}{C_{00}}\sqrt{C_{22}C_{00} - C_{02}^2} \right], \tag{26}$$

In the full equations above, the notation $I_3(i, k, j)$ maps to $x_0 = \lambda_i$, $x_1 = \lambda_k$, and $x_2 = \lambda_j$.

### A.5. Loss Function in Terms of Order Parameters

Finally, utilising the definition of the two-variable integral $I_2$, we can express the population loss $\mathcal{L}$ in a compact closed form. Expanding the squared error term and exchanging the summation and expectation operations, we obtain:

$$\mathcal{L} = \frac{1}{2}\left\langle \left( \sum_{k=1}^{K} v_k g(\lambda_k) - \sum_{m=1}^{M} v_m^* g(\rho_m) \right)^2 \right\rangle$$

$$= \frac{1}{2}\sum_{i,j=1}^{K} v_i v_j \langle g(\lambda_i)g(\lambda_j) \rangle + \frac{1}{2}\sum_{n,m=1}^{M} v_n^* v_m^* \langle g(\rho_n)g(\rho_m) \rangle - \sum_{k=1}^{K}\sum_{m=1}^{M} v_k v_m^* \langle g(\lambda_k)g(\rho_m) \rangle. \tag{27}$$

Substituting the integral definition $I_2(x, y) = \langle g(x)g(y) \rangle$, the loss function becomes:

$$\mathcal{L} = \frac{1}{2}\sum_{i,j=1}^{K} v_i v_j I_2(i, j) + \frac{1}{2}\sum_{n,m=1}^{M} v_n^* v_m^* I_2(n, m) - \sum_{k=1}^{K}\sum_{m=1}^{M} v_k v_m^* I_2(k, m). \tag{28}$$

### A.6. Stochastic Gradient Descent

In the case of Stochastic Gradient Descent, as shown in Saad & Solla (1995); Biehl & Schwarze (1995), the ODE for $Q$ presents an additional term resulting from the stochasticity of the update. This results in the equations below:

$$\frac{dR_{in}}{dt} = \eta v_i \left[ \sum_{m=1}^{M} v_m^* I_3(i, n, m) - \sum_{j=1}^{K} v_j I_3(i, n, j) \right], \tag{29}$$

$$\frac{dQ_{ik}}{dt} = \eta v_i \left[ \sum_{m=1}^{M} v_m^* I_3(i, k, m) - \sum_{j=1}^{K} v_j I_3(i, k, j) \right] + \eta v_k \left[ \sum_{m=1}^{M} v_m^* I_3(k, i, m) - \sum_{j=1}^{K} v_j I_3(k, i, j) \right]$$

$$+ \eta^2 v_i v_k \left[ \sum_{m,n=1}^{M} v_m^* v_n^* I_4(i, k, n, m) - 2\sum_{j=1}^{K}\sum_{m=1}^{M} v_j v_m^* I_4(i, k, m, j) + \sum_{j,l=1}^{K} v_j v_l I_4(i, k, j, l) \right], \tag{30}$$

$$\frac{dv_i}{dt} = \eta \left[ \sum_{n=1}^{M} v_n^* I_2(i, n) - \sum_{j=1}^{K} v_j I_2(i, j) \right]. \tag{31}$$

Where the new term $I_4$ is given by $I_4(x_0, x_1, x_2, x_3) = \langle g'(x_0)\, g'(x_1)\, g(x_2)\, g(x_3) \rangle$.

## B. Equations of Fixed Points

The implicit equations described in Result 1 describing the fixed points are the following:

$$0 = \mathcal{F}_R(Q, R) = \frac{dR_{in}}{dt} = \sum_{m=1}^{M} I_3(i, n, m) - \sum_{j=1}^{K} I_3(i, n, j),$$

$$0 = \mathcal{F}_Q(Q,R) = \frac{dQ_{ik}}{dt} = \sum_{m=1}^{M} I_3(i,k,m) - \sum_{j=1}^{K} I_3(i,k,j) + \sum_{m=1}^{M} I_3(k,i,m) - \sum_{j=1}^{K} I_3(k,i,j),$$

where $I_3$ for the case of ReLU is provided in the Appendix Eq.26.

These equations are derived from the gradient flow dynamics. It is easy to see, and discuss in detail in Appendix A, that the equations are described by the order paramters Eqs.5 introduced in the Main Text. Since the gradient depends on the weights only through their overlaps $Q$ and $R$, any stationary point of the high-dimensional population risk Eq.27 is necessarily a stationary point of the summary statistics equations $\mathcal{F}_R(Q,R) = 0$, $\mathcal{F}_Q(Q,R) = 0$ for Gaussian inputs, therefore the flow must be zero giving Result 1.

As discussed in Sec.3.2, our results are more general than just gradient flow on two-layer ReLU networks. In Appendix A, F, and E we discuss how the flow is derive more in general, including other activation function and the stochastic gradient dynamics case.

### B.1. Ansatz Formulation

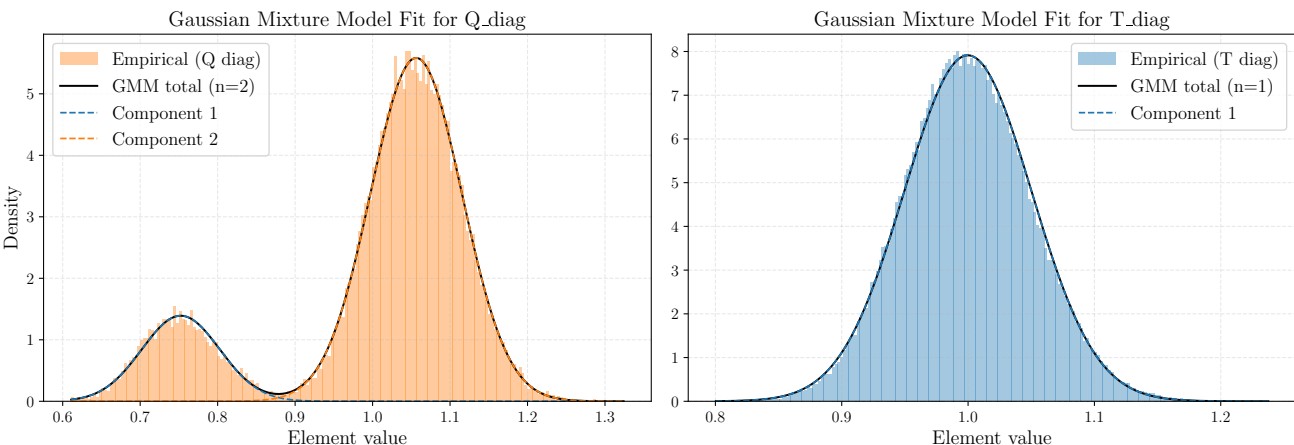

*Figure 6.* **Gaussian mixture analysis of diagonal order parameters.** The left panel shows the distribution of diagonal elements of the matrix $Q$, which is well described by a two-component Gaussian mixture model. The right panel shows the distribution of diagonal elements of the matrix $T$, which follows a single Gaussian distribution centred at 1. The data are obtained by grouping order parameters within the same minima family, re-arranging the corresponding $(Q,R)$ matrices to maximise the maximise the absolute values of diagonal of $Q$ and extracting the diagonal entries.

To characterise the structure of the order parameters, we employ a Gaussian Mixture Model (GMM) analysis, which represents a distribution as a weighted sum of Gaussian components and provides a simple tool for identifying clustered structure in empirical data. We focus on the family of minima empirically observed by the results from large-scale cluster runs (e.g., the same structure shown in Fig.1b). In order to show the order parameter matrices $R$ and $Q$ in a consistent way, we have to consider the permutational invariance of the units in the solution. Therefore, we re-arrange the student's units to maximise the absolute values of the diagonal elements of the student-teacher overlap $R$.

After alignment, we analyse the distribution of the diagonal elements of the matrices $Q$ and $T$ across the ensemble of minima by fitting these distributions using a GMM to identify distinct populations of hidden units.

In Fig.6, we show an example of this procedure for elements in $Q$. In Fig. 6 (Left), the diagonal elements of $Q$ exhibit a clear bimodal distribution. A two-component GMM reveals two well-separated Gaussian components, whose relative weights closely match the ratio $k_1/k_2$ (with $k_2$ denoting the size of the aligned group). This indicates that student units separate into two distinct populations during training, supporting the block-structured ansatz used in our theoretical analysis. As a control, in Fig.6 (Right) we apply the same GMM analysis to the diagonal elements of the teacher-teacher overlap matrix $T$. In contrast to $Q$, the distribution of $T$ is well described by a single Gaussian component ($n = 1$), as shown in Fig. 6 (Right).

This intuition is used to derive the ansatz reported in Sec.3.1.

The block-symmetric ansatz proposed in Eq. 7 can be formally related to the discrete group-theoretic analysis of Arjevani &

Field (2019). They demonstrate that critical points in two-layer ReLU networks under Gaussian inputs retain large isotropy subgroups (e.g., $S_{k_1} \times S_{k_2}$), leading to weight matrices with highly symmetric transitivity partitions.

In our macroscopic framework, these discrete weight-space partitions manifest exactly as the uniform block-diagonal and off-diagonal constants in the $Q$ and $R$ order parameters. Therefore, the dynamical compensation mechanism we observe is the statistical-mechanics dual to their "principle of least symmetry breaking."

**Examples of the Ansatz.** We report here some examples for the case $k_1 = 2, k_2 = 3$ in the well-specified regime $(K = M)$:

$$R = \begin{pmatrix} \varsigma & s & \varphi & \varphi & \varphi \\ s & \varsigma & \varphi & \varphi & \varphi \\ f & f & b & \tau & \tau \\ f & f & \tau & b & \tau \\ f & f & \tau & \tau & b \end{pmatrix}, \quad Q = \begin{pmatrix} q & e & u & u & u \\ e & q & u & u & u \\ u & u & p & \mu & \mu \\ u & u & \mu & p & \mu \\ u & u & \mu & \mu & p \end{pmatrix}. \tag{32}$$

and over-parametrised case (i.e. $K = M + 1$):

$$R = \begin{pmatrix} \iota & \iota & \kappa & \kappa & \kappa \\ \varsigma & s & \varphi & \varphi & \varphi \\ s & \varsigma & \varphi & \varphi & \varphi \\ f & f & b & \tau & \tau \\ f & f & \tau & b & \tau \\ f & f & \tau & \tau & b \end{pmatrix}, \quad Q = \begin{pmatrix} \Omega & v & v & \gamma & \gamma & \gamma \\ v & q & e & u & u & u \\ v & e & q & u & u & u \\ \gamma & u & u & p & \mu & \mu \\ \gamma & u & u & \mu & p & \mu \\ \gamma & u & u & \mu & \mu & p \end{pmatrix}. \tag{33}$$

To provide a direct visual check of the structural assumptions underlying the ansatz, we report a representative example of the order parameters $(R^*, Q^*)$ obtained from the simulations for $K = 18$ and $M = 17$, shown in Fig. 7.

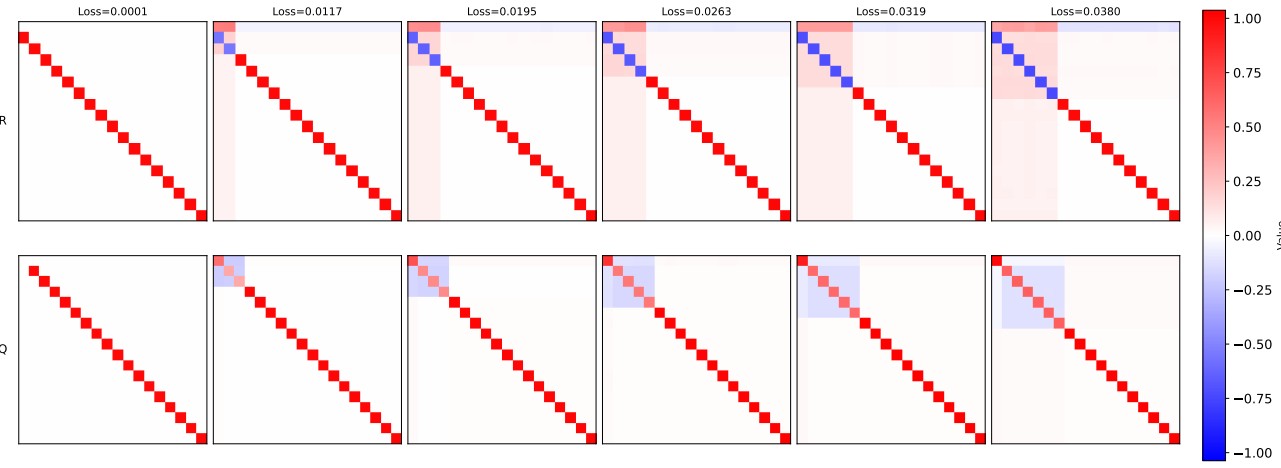

*Figure 7.* **Different order parameters at $K = 18$ and $M = 17$.** The plots show the values of $R$ (first row) and $Q$ (second row) for minima representative of the different families. From left to right, we see results for $k_1 = 0, 2, 3, 4, 5, 6$.

### B.2. Solution of the Fixed Point Equation Under the Ansatz

The fixed point equations of $dQ_{ik}/dt$, $dR_{in}/dt$ (Eqs. 21 and Eqs. 20) simplify after substituting the ansatz defined in Eqs. 7. Different entries of $Q_{ik}$ and $R_{in}$ (e.g., $e$ and $\varsigma$) satisfy distinct equations, denoted by $E_e$, $E_\varsigma$, and so on. In the $K = M$ setting, this procedure results in 11 coupled equations characterising fixed points reported below:

$$E_\varsigma = -\frac{1}{2\pi q}[k_1 \varsigma \sqrt{q^2 - e^2} - \varsigma \sqrt{q^2 - e^2} + (k_1 - 1) q s \cos^{-1}\left(-\frac{e}{\sqrt{q^2}}\right) + f k_2 q \cos^{-1}\left(-\frac{u}{\sqrt{pq}}\right) - k_1 \varsigma \sqrt{q - s^2}$$

$$+ k_2 \varsigma \sqrt{pq - u^2} - k_2 \varsigma \sqrt{q - \phi^2} + q \varsigma \cos^{-1}\left(-\frac{q}{\sqrt{q^2}}\right) + \varsigma \sqrt{q - s^2} - \varsigma \sqrt{q - \varsigma^2} - q \cos^{-1}\left(-\frac{\varsigma}{\sqrt{q}}\right)] = 0, \tag{34}$$

$$E_s = -\frac{1}{2\pi q}[k_1\, s\sqrt{q^2 - e^2} - s\sqrt{q^2 - e^2} + q\cos^{-1}\left(-\frac{e}{\sqrt{q^2}}\right)\left((k_1 - 2)\,s + \varsigma\right) + f\,k_2\,q\cos^{-1}\left(-\frac{u}{\sqrt{pq}}\right) - k_1\,s\sqrt{q - s^2}$$

$$+ k_2\,s\sqrt{pq - u^2} - k_2\,s\sqrt{q - \varphi^2} + q\,s\cos^{-1}\left(-\frac{q}{\sqrt{q^2}}\right) + s\sqrt{q - s^2} - s\sqrt{q - \varsigma^2} - q\cos^{-1}\left(-\frac{s}{\sqrt{q}}\right)] = 0, \tag{35}$$

$$E_\varphi = \frac{1}{2\pi q}[-q\big(b + (k_2 - 1)\tau\big)\cos^{-1}\left(-\frac{u}{\sqrt{pq}}\right) + \varphi\left(-(k_1 - 1)\left(\sqrt{q^2 - e^2} + q\cos^{-1}\left(-\frac{e}{\sqrt{q^2}}\right)\right) - q\cos^{-1}\left(-\frac{q}{\sqrt{q^2}}\right)\right)$$

$$+ (k_1 - 1)\,\varphi\sqrt{q - s^2} - k_2\,\varphi\sqrt{pq - u^2} + k_2\,\varphi\sqrt{q - \varphi^2} + \varphi\sqrt{q - \varsigma^2} + q\cos^{-1}\left(-\frac{\varphi}{\sqrt{q}}\right)] = 0, \tag{36}$$

$$E_e = \frac{1}{\pi q}\Big[k_1\left(e\left(\sqrt{q - s^2} - \sqrt{q^2 - e^2}\right) + qs\cos^{-1}\left(-\frac{s}{\sqrt{q}}\right)\right) + e\sqrt{q^2 - e^2} - q\big(e(k_1 - 2) + q\big)\cos^{-1}\left(-\frac{e}{\sqrt{q^2}}\right)$$

$$+ k_2\left(e\left(\sqrt{q - \varphi^2} - \sqrt{pq - u^2}\right) - qu\cos^{-1}\left(-\frac{u}{\sqrt{pq}}\right)\right) - eq\cos^{-1}\left(-\frac{q}{\sqrt{q^2}}\right) - e\sqrt{q - s^2} + e\sqrt{q - \varsigma^2} \tag{37}$$

$$+ k_2\,\varphi\,q\cos^{-1}\left(-\frac{\varphi}{\sqrt{q}}\right) + q\,\varsigma\cos^{-1}\left(-\frac{s}{\sqrt{q}}\right) + qs\cos^{-1}\left(-\frac{\varsigma}{\sqrt{q}}\right) - 2qs\cos^{-1}\left(-\frac{s}{\sqrt{q}}\right)\Big] = 0,$$

$$E_u = \frac{1}{2\pi}\Bigg[\frac{u\sqrt{p - b^2}}{p} + \varphi\cos^{-1}\left(-\frac{b}{\sqrt{p}}\right) + b\cos^{-1}\left(-\frac{\varphi}{\sqrt{q}}\right) - u\left(\frac{(k_1 - 1)\left(\sqrt{q^2 - e^2} + q\cos^{-1}\left(-\frac{e}{\sqrt{q^2}}\right)\right)}{q} + \cos^{-1}\left(-\frac{q}{\sqrt{q^2}}\right)\right)$$

$$- \frac{1}{p}\left(p\big(e(k_1 - 1) + q\big)\cos^{-1}\left(-\frac{u}{\sqrt{pq}}\right) + k_1 u\sqrt{pq - u^2}\right) + (k_1 - 1)\left(\frac{u\sqrt{p - f^2}}{p} + s\cos^{-1}\left(-\frac{f}{\sqrt{p}}\right)\right)$$

$$+ \frac{u\sqrt{p - f^2}}{p} + (k_1 - 1)\left(f\cos^{-1}\left(-\frac{s}{\sqrt{q}}\right) + \frac{u\sqrt{q - s^2}}{q}\right) + \varsigma\cos^{-1}\left(-\frac{f}{\sqrt{p}}\right) + f\cos^{-1}\left(-\frac{\varsigma}{\sqrt{q}}\right)$$

$$- u\left(\frac{(k_2 - 1)\left(\sqrt{p^2 - \mu^2} + p\cos^{-1}\left(-\frac{\mu}{\sqrt{p^2}}\right)\right)}{p} + \cos^{-1}\left(-\frac{p}{\sqrt{p^2}}\right)\right) - \frac{q\big((k_2 - 1)\mu + p\big)\cos^{-1}\left(-\frac{u}{\sqrt{pq}}\right) + k_2 u\sqrt{pq - u^2}}{q}$$

$$+ (k_2 - 1)\left(\varphi\cos^{-1}\left(-\frac{\tau}{\sqrt{p}}\right) + \frac{u\sqrt{p - \tau^2}}{p}\right) + (k_2 - 1)\left(\frac{u\sqrt{q - \varphi^2}}{q} + \tau\cos^{-1}\left(-\frac{\varphi}{\sqrt{q}}\right)\right) + \frac{u\sqrt{q - \varphi^2}}{q} + \frac{u\sqrt{q - \varsigma^2}}{q}\Bigg] = 0, \tag{38}$$

$$E_f = -\frac{1}{2\pi p} - f\sqrt{p - b^2} - fk_1\sqrt{p - f^2} + fk_1\sqrt{pq - u^2} + fk_2\sqrt{p^2 - \mu^2} + f(k_2 - 1)\,p\cos^{-1}\left(-\frac{\mu}{\sqrt{p^2}}\right) - fk_2\sqrt{p - \tau^2}$$

$$- f\sqrt{p^2 - \mu^2} + fp\cos^{-1}\left(-\frac{p}{\sqrt{p^2}}\right) + f\sqrt{p - \tau^2} - p\cos^{-1}\left(-\frac{f}{\sqrt{p}}\right) + k_1 ps\cos^{-1}\left(-\frac{u}{\sqrt{pq}}\right) - ps\cos^{-1}\left(-\frac{u}{\sqrt{pq}}\right)$$

$$+ p\varsigma\cos^{-1}\left(-\frac{u}{\sqrt{pq}}\right) = 0, \tag{39}$$

$$E_b = -\frac{1}{2\pi p}[-b\sqrt{p - b^2} - bk_1\sqrt{p - f^2} + bk_1\sqrt{pq - u^2} + bk_2\sqrt{p^2 - \mu^2} - bk_2\sqrt{p - \tau^2} - b\sqrt{p^2 - \mu^2} + bp\cos^{-1}\left(-\frac{p}{\sqrt{p^2}}\right)$$

$$+ b\sqrt{p - \tau^2} - p\cos^{-1}\left(-\frac{b}{\sqrt{p}}\right) + k_1 p\varphi\cos^{-1}\left(-\frac{u}{\sqrt{pq}}\right) + (k_2 - 1)p\tau\cos^{-1}\left(-\frac{\mu}{\sqrt{p^2}}\right)] = 0, \tag{40}$$

$$E_\tau = -\frac{1}{2\pi p}[-\tau\sqrt{p-b^2} + p(b+(k_2-2)\tau)\cos^{-1}\left(-\frac{\mu}{\sqrt{p^2}}\right) - k_1\tau\sqrt{p-f^2} + k_1 p\varphi\cos^{-1}\left(-\frac{u}{\sqrt{pq}}\right) + k_1\tau\sqrt{pq-u^2}$$

$$+ k_2\tau\sqrt{p^2-\mu^2} - k_2\tau\sqrt{p-\tau^2} - \tau\sqrt{p^2-\mu^2} + p\tau\cos^{-1}\left(-\frac{p}{\sqrt{p^2}}\right) + \tau\sqrt{p-\tau^2} - p\cos^{-1}\left(-\frac{\tau}{\sqrt{p}}\right)] = 0, \tag{41}$$

$$E_q = \frac{1}{\pi}[-(k_1-1)\sqrt{q^2-e^2} - e(k_1-1)\cos^{-1}\left(-\frac{e}{\sqrt{q^2}}\right) + (k_1-1)\left(\sqrt{q-s^2} + s\cos^{-1}\left(-\frac{s}{\sqrt{q}}\right)\right)$$

$$- k_2\left(\sqrt{pq-u^2} + u\cos^{-1}\left(-\frac{u}{\sqrt{pq}}\right)\right) + k_2\left(\sqrt{q-\varphi^2} + \varphi\cos^{-1}\left(-\frac{\varphi}{\sqrt{q}}\right)\right) - q\cos^{-1}\left(-\frac{q}{\sqrt{q^2}}\right) + \sqrt{q-\varsigma^2} \tag{42}$$

$$+ \varsigma\cos^{-1}\left(-\frac{\varsigma}{\sqrt{q}}\right)] = 0,$$

$$E_p = \frac{1}{\pi}[\sqrt{p-b^2} + b\cos^{-1}\left(-\frac{b}{\sqrt{p}}\right) + k_1\left(\sqrt{p-f^2} + f\cos^{-1}\left(-\frac{f}{\sqrt{p}}\right)\right) - k_1\left(\sqrt{pq-u^2} + u\cos^{-1}\left(-\frac{u}{\sqrt{pq}}\right)\right)$$

$$- (k_2-1)\sqrt{p^2-\mu^2} + (k_2-1)(-\mu)\cos^{-1}\left(-\frac{\mu}{\sqrt{p^2}}\right) + k_2\sqrt{p-\tau^2} + (k_2-1)\tau\cos^{-1}\left(-\frac{\tau}{\sqrt{p}}\right) - p\cos^{-1}\left(-\frac{p}{\sqrt{p^2}}\right)$$

$$- \sqrt{p-\tau^2}] = 0, \tag{43}$$

$$E_\mu = \frac{1}{\pi p}\Big[\mu\sqrt{p-b^2} + p\tau\cos^{-1}\left(-\frac{b}{\sqrt{p}}\right) + bp\cos^{-1}\left(-\frac{\tau}{\sqrt{p}}\right) + k_1\left(\mu(\sqrt{p-f^2} - \sqrt{pq-u^2}) + fp\cos^{-1}\left(-\frac{f}{\sqrt{p}}\right)\right.$$

$$\left. -pu\cos^{-1}\left(-\frac{u}{\sqrt{pq}}\right)\right) - k_2\mu\left(\sqrt{p^2-\mu^2} + p\cos^{-1}\left(-\frac{\mu}{\sqrt{p^2}}\right)\right) + k_2\mu\sqrt{p-\tau^2} + k_2 p\tau\cos^{-1}\left(-\frac{\tau}{\sqrt{p}}\right)$$

$$+ \mu\sqrt{p^2-\mu^2} - p^2\cos^{-1}\left(-\frac{\mu}{\sqrt{p^2}}\right) + 2\mu p\cos^{-1}\left(-\frac{\mu}{\sqrt{p^2}}\right) - \mu p\cos^{-1}\left(-\frac{p}{\sqrt{p^2}}\right) \tag{44}$$

$$- \mu\sqrt{p-\tau^2} - 2p\tau\cos^{-1}\left(-\frac{\tau}{\sqrt{p}}\right)\Big] = 0.$$

To validate the theoretical ansatz proposed in the Main Text, we solve the reduced fixed-point equations derived above. Specifically, we substitute the block-diagonal parameterization into the general mean-field ODEs (derived in Appendix A.3), transforming the high-dimensional optimization problem into a set of coupled non-linear algebraic equations. We used the solver "FindRoot" implemented in Wolfram Mathematica to solve the fixed point equations. The function implements a Newton's method implementation. The resulting solutions $(R^*, Q^*)$ constitute the theoretical fixed points of the mean-field dynamics. By evaluating the population loss function at these theoretical coordinates, we obtain the predicted loss values. These values correspond to the vertical dashed lines shown in Fig. 2 (Main Text), which demonstrate a close agreement with the peaks of the empirical loss histograms.

Following the same procedure as in the case $K = M$, the above analysis can be straightforwardly generalized to the overparameterised setting $K = M + 1$. Due to their length, we do not report these equations explicitly. Instead, Fig. 8 displays the order parameters $(R^*, Q^*)$ obtained directly from the numerical solver for $K = 18$ and $M = 17$ in different $k_1$. The resulting structures and losses closely match those shown in Fig. 7.

## C. Robustness to Finite Input Dimensionality

In the main text, our analysis of the fixed points we assume orthonormal teacher configuration $(T = I_M)$ for simplifying the equations. While this is exact in the thermodynamic limit $(d \to \infty)$, in this section, we investigate the robustness of our predictions to finite input dimensions $d$. We simulate the dynamics starting from actual neural network weight initialisations at dimensions $d \in \{196, 392, 784\}$.

Fig. 9 illustrates the impact of finite dimensionality on the final population loss distribution. We compare the empirical histograms obtained from finite $d$ simulations against the idealised infinite-dimensional case. While the discrete, quantised hierarchy of the local minima is strictly preserved, finite dimensions introduce variance. This results in a progressive broadening of the loss distribution peaks around the theoretically predicted values. As expected, the empirical distributions tightly converge toward the analytical infinite-dimensional limit as $d$ increases.

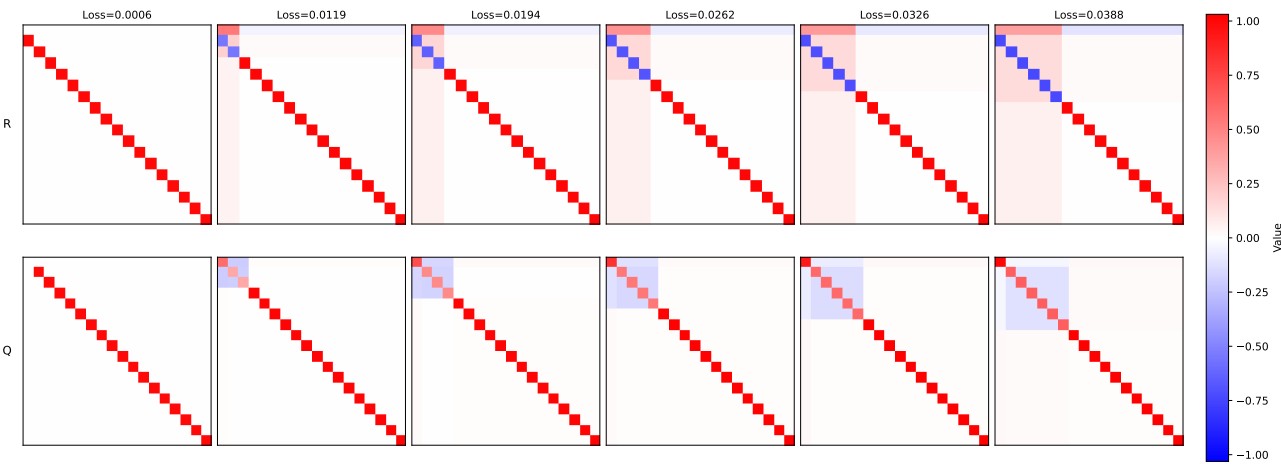

*Figure 8.* **Order parameters obtained from solvers at $K = 18$ and $M = 17$.** The plots show the values of $R$ (first row) and $Q$ (second row) for minima computed by the solver. Columns are ordered from left to right by the number of anti-aligned units, $k_1 = 0, 2, 3, 4, 5, 6$.

To further validate that the structural symmetries discussed in our ansatz (Sec.3.1) persist beyond the idealised orthonormal teacher scenario, Fig.10 visualises the student-teacher overlap matrix $R$. We uniformly sample 10 configurations at the end of the dynamics from the first-order local minimum (i.e., the second quantised group corresponding to $k_1 = 1$, as observed in Fig.9). Despite the noise introduced by the finite-dimensional teacher, the distinct macroscopic block structure—including the aligned and anti-aligned unit clusters—remains clearly identifiable. As $d$ decreases, the off-diagonal fluctuations become more pronounced, yet the fundamental geometric organisation in blocks remains robust. This confirms that our ansatz accurately reflects the empirical behaviour of finite-dimensional networks.

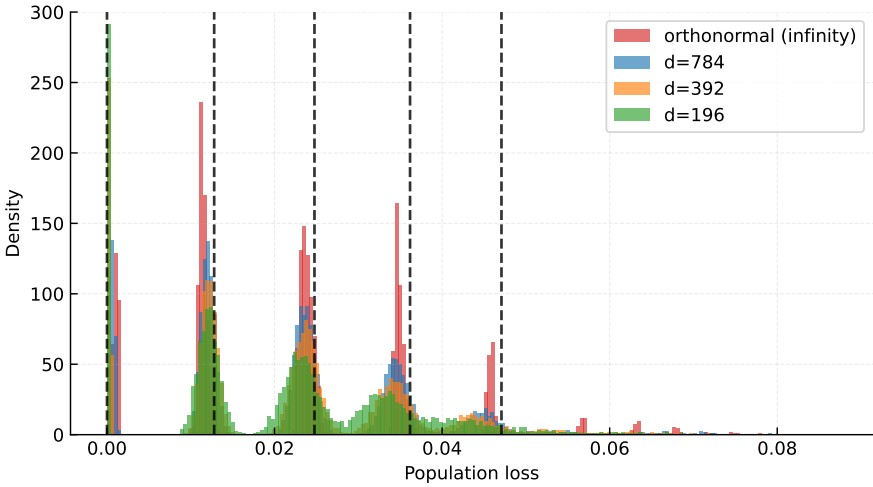

*Figure 9.* **Impact of finite dimensionality on the loss distribution.** Histograms of the final population loss obtained by simulating the ODEs starting from actual neural network weight initialisations at dimensions $d \in \{196, 392, 784\}$, compared against the infinite-dimensional limit (orthonormal teacher, $T = \mathbb{I}$). Dashed vertical lines indicate the corresponding theoretically predicted loss values. While smaller dimensions introduce a broadening effect due to variance, the discrete families of minima remain robustly centred around the predicted macroscopic levels.

## D. Stability of fixed points from higher order minima

As noted in the Main Text, the non-differentiability of the ReLU activation function at the origin precludes a standard Hessian-based stability analysis. We employ a perturbative approach to probe the local landscape geometry.

Given a fixed point configuration $\bar{W}$ derived by solving the fixed point equations in the well-specified setting (Appendix B.2),

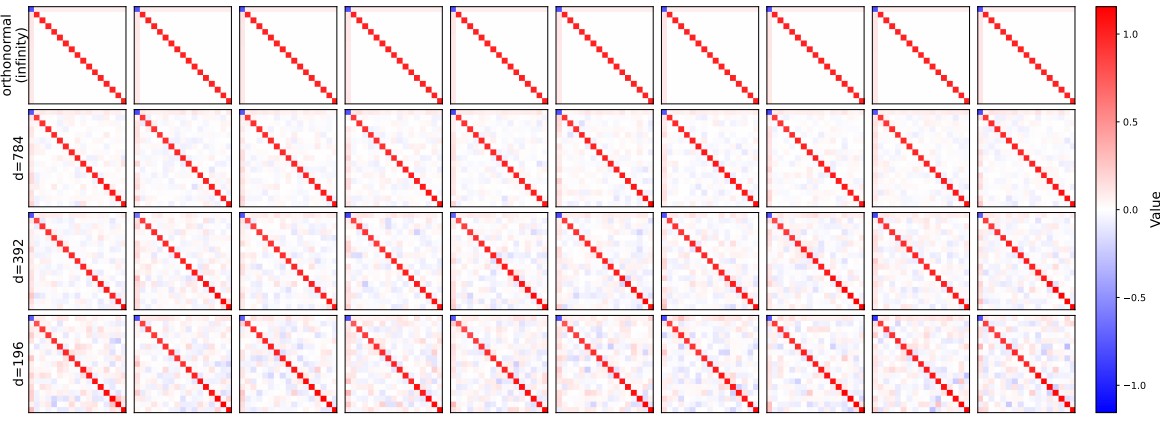

*Figure 10.* **Student-teacher overlap matrix ($R$) across different dimensions.** Heatmaps of the $R$ matrix for 10 randomly sampled configurations converging to the first-order local minimum ($k_1 = 1$). The top row represents the idealised $d \to \infty$ limit, while subsequent rows correspond to finite dimensions $d = 784, 392$, and $196$. The block-symmetric structure predicted by our theoretical ansatz clearly emerges across all regimes, with finite-size fluctuations increasing at lower dimensions.

we apply an isotropic Gaussian perturbation in the weight space to see if it acts as a stable attractor or a saddle point:

$$W_{\text{init}} = \bar{W} + \xi, \quad \text{with} \quad \xi \sim \mathcal{N}(0, \sigma^2 I) \tag{45}$$

where $\sigma$ represents the strength of the perturbation.

The perturbation maps into equivalent perturbation of $Q$ and $R$ that are then analysed in the mean-field description using $Q$ and $R$. Indeed, directly perturbing $Q$ and $R$ may result in non-realisable configurations, in particular $Q \succeq 0$, $|Q_{ij}| = |Q_{ji}| \leq \sqrt{Q_{ii}Q_{jj}}$, and $|R_{im}| \leq \sqrt{Q_{ii}T_{mm}}$.

The system is then allowed to relax for $10,000$ steps following the gradient flow dynamics with $\eta = 0.01$. We monitor the discrepancy in loss value between the unperturbed configuration and the perturbed one.

The perturbation is repeated 20 times with different seeds, and the average and the variance of the loss gap are recorded and shown in Fig.3 of the Main Text.

Our empirical observation that spurious fixed points become unstable for $K \geq M + 1$ is supported by the exact local Hessian analysis of Safran et al. (2021). They mathematically proved that taking a non-global minimum in the $K = M$ landscape and "splitting" a neuron to embed it in the $K = M + 1$ landscape necessarily introduces a strictly negative eigenvalue in the Hessian block corresponding to the duplicated units.

When we introduce the Gaussian perturbation $\xi$ to the extended weight matrix, the system is pushed off the exact critical point. Because the Hessian now possesses a direction of negative curvature, the gradient flow does not return to the spurious state, but instead escapes the newly formed saddle point and descends toward lower-loss configurations.

## E. Alternative Activation Functions

In this section, we provide the analytical expressions for the Gaussian integrals $I_2$ and $I_3$ for Leaky ReLU and Error Function (erf) activations. Based on these derivations, we also present numerical observations of the learning landscape and spurious minima structures for these activations in this section. The notation follows the definitions in Appendix A:

$$I_2(x_1, x_2) = \langle g(x_1)g(x_2)\rangle, \tag{46}$$

$$I_3(x_0, x_1, x_2) = \langle g'(x_0)x_1 g(x_2)\rangle, \tag{47}$$

where the expectations are taken over a multivariate Gaussian distribution with covariance matrix $\Sigma$.

### E.1. Leaky ReLU

The Leaky ReLU activation function with a leakage parameter $\alpha \in [0, 1]$ is defined as:

$$g(x) = \text{LReLU}(x; \alpha) = \begin{cases} x & \text{if } x \geq 0, \\ \alpha x & \text{if } x < 0. \end{cases} \tag{48}$$

Its derivative is the generalized step function $g'(x) = \mathbb{I}(x \geq 0) + \alpha\mathbb{I}(x < 0)$. Note that setting $\alpha = 0$ recovers the standard ReLU results, while $\alpha = 1$ corresponds to a linear network.

**Two-Variable Case**  The explicit form for the correlation between two Leaky ReLU units is:

$$I_2^{\text{LReLU}}(\Sigma) = \alpha C_{12} + \frac{(1-\alpha)^2}{2\pi}\left[\sqrt{C_{11}C_{22} - C_{12}^2} + C_{12}\arccos\left(-\frac{C_{12}}{\sqrt{C_{11}C_{22}}}\right)\right], \tag{49}$$

where $C_{11}, C_{22}$ are the variances and $C_{12}$ is the covariance of $(x_1, x_2)$.

**Three-Variable Case**  The integral involving the derivative, a multiplier, and the activation is given by:

$$I_3^{\text{LReLU}}(\Sigma) = \alpha C_{12} + \frac{(1-\alpha)^2}{2\pi}\left[C_{12}\arccos\left(-\frac{C_{02}}{\sqrt{C_{22}C_{00}}}\right) + \frac{C_{01}}{C_{00}}\sqrt{C_{22}C_{00} - C_{02}^2}\right]. \tag{50}$$

Here, the covariance matrix for the joint variables $(x_0, x_1, x_2)$ is:

$$\Sigma = \begin{pmatrix} C_{00} & C_{01} & C_{02} \\ C_{01} & C_{11} & C_{12} \\ C_{02} & C_{12} & C_{22} \end{pmatrix}.$$

### E.2. Leaky ReLU Results

Based on the derived integrals, we numerically investigate the stationary points of networks with leaky ReLU activation. With the Leaky ReLU parameter set to $\alpha = 0.01$, the Fig. 11 shows the results for $K = M = 12$ under nGD. It can be clearly observed that anti-aligned units appear, similar to the ReLU case.

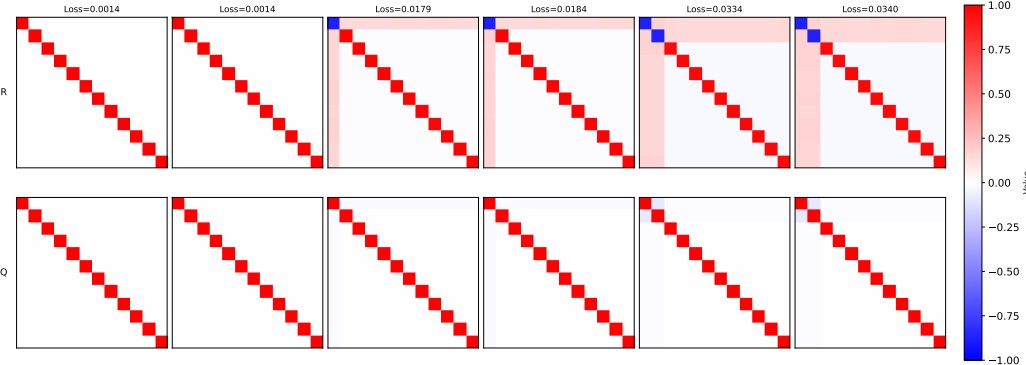

*Figure 11.* Order Parameters of Leaky ReLU in $K = M = 12$

### E.3. Sigmoidal (erf)

Consider the sigmoidal activation function defined by the error function:

$$g(x) = \text{erf}\left(\frac{x}{\sqrt{2}}\right). \tag{51}$$

The derivative is proportional to a Gaussian: $g'(x) = \sqrt{\frac{2}{\pi}}e^{-x^2/2}$.

**Two-Variable Case ($I_2$)**  Based on the arcsine law for Gaussian integrals, the result is:

$$I_2^{\text{erf}}(\Sigma) = \frac{2}{\pi} \arcsin\left(\frac{C_{12}}{\sqrt{(1+C_{11})(1+C_{22})}}\right). \tag{52}$$

**Three-Variable Case ($I_3$)**  The integral involving the derivative of the error function allows for an analytical solution:

$$I_3^{\text{erf}}(\Sigma) = \frac{2}{\pi\sqrt{(1+C_{00})(1+C_{22})-C_{02}^2}}\left(C_{12} - \frac{C_{01}C_{02}}{1+C_{00}}\right). \tag{53}$$

### E.4. Results for Sigmoidal (erf) Activation

Based on the derived integrals, we numerically investigate the stationary points of networks with erf activation by running the ODEs for a few seeds. Fig. 12 illustrates the structure of the order parameters for $K = M = 12$ across different local minima found by nGD. Notice the absence of anti-aligned neurons in sharp contrast with what is observed in the Main Text for ReLU and in Fig.11 for Leaky ReLU.

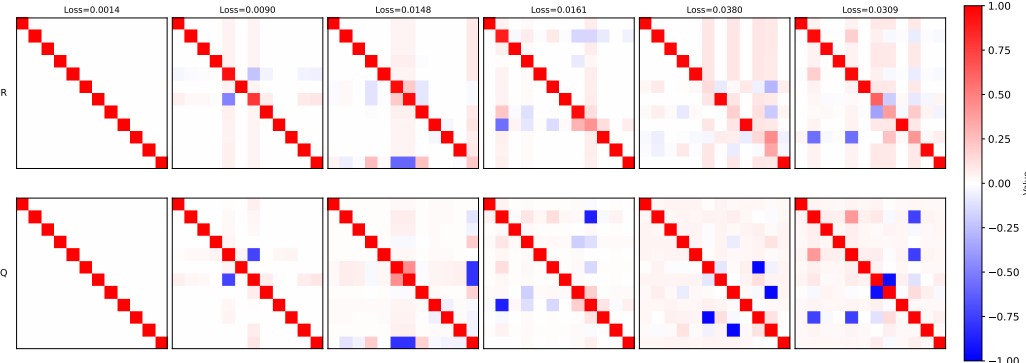

*Figure 12.* Order Parameters of erf in $K = M = 12$

## F. Constraint Dynamics

In this section, we derive the mean-field ODEs for Normalised Gradient Descent (nGD) and Orthonormalized Gradient Descent (onGD), which impose distinct geometric constraints on the student weights during training. We also show the resulting learning dynamics for both methods and a brief discussion of stability, convergence properties, and the learnability of nGD and onGD. Consistent with previous sections, we denote the input dimension by $d$.

### F.1. Normalised Student Setting (nGD)

In Normalised Gradient Descent, we enforce the constraint that the norm of each student weight vector remains fixed at initialisation, typically $\|w_i\|^2 = d$ (implying $Q_{ii} = 1$). The continuous-time update rule subtracts the radial component of the gradient:

$$\frac{dw_i}{dt} = -\eta \nabla_{w_i}\mathcal{L} + \frac{\eta}{d}(\nabla_{w_i}\mathcal{L} \cdot w_i^\top)w_i. \tag{54}$$

This projection modifies the order parameter dynamics by introducing a decay term proportional to the self-overlap. The resulting ODEs are:

$$\frac{dR_{in}}{dt} = \eta v_i \left[\sum_{m=1}^{M} v_m^* I_3(i,n,m) - \sum_{j=1}^{K} v_j I_3(i,n,j)\right] - \eta R_{in} v_i \left[\sum_{m=1}^{M} v_m^* I_3(i,i,m) - \sum_{j=1}^{K} v_j I_3(i,i,j)\right], \tag{55}$$

$$\frac{dQ_{ik}}{dt} = \eta v_i \left[\sum_{m=1}^{M} v_m^* I_3(i,k,m) - \sum_{j=1}^{K} v_j I_3(i,k,j)\right] + \eta v_k \left[\sum_{m=1}^{M} v_m^* I_3(k,i,m) - \sum_{j=1}^{K} v_j I_3(k,i,j)\right]$$

$$-\eta Q_{ik}v_i\left[\sum_{m=1}^{M} v_m^* I_3(i,i,m) - \sum_{j=1}^{K} v_j I_3(i,i,j)\right] - \eta Q_{ik}v_k\left[\sum_{m=1}^{M} v_m^* I_3(k,k,m) - \sum_{j=1}^{K} v_j I_3(k,k,j)\right]. \quad (56)$$

Note that for the diagonal elements, substituting $k = i$ and $Q_{ii} = 1$ confirms that $\frac{dQ_{ii}}{dt} = 0$, satisfying the constraint.

### F.2. Empirical Observations for Normalized Gradient Descent

We investigate the loss distribution of Normalized Gradient Descent (nGD) in the overparameterised regime. Specifically, we consider a student network with $K = 19$ hidden units learning from a teacher with $M = 17$ units ($K = M + 2$). Fig. 13a displays the histogram of the final population loss.

Contrary to the expectation that overparameterisation allows the network to achieve zero loss, nGD exhibits a fundamental obstruction. As shown in the histogram, the system fails to reach the rigorous global minima (Loss $\approx 0$). Even in the best-found minima (Loss $\approx 0.355$), a significant error persists.

This is a direct consequence of the constraint $Q_{ii} = 1$. In unconstrained GD, the $K - M$ excess students would decay to zero norm to eliminate interference, as shown in Fig. 7. In nGD, however, the order parameters shown in Fig. 14 (left columns) indicate that these excess units are constrained to maintain unit norm. They cannot be silenced and effectively act as intrinsic noise sources, creating an irreducible error even when the $M$ target features are perfectly retrieved.

The high density of solutions in the suboptimal interval Loss $\in [0.365, 0.370]$ (approx. $48\%$) arises from the combinatorial diversity of "blocking" defects. As visualised in the right four columns of Fig. 14, these states exhibit student groupings of varying sizes (e.g., $k_1 = 3, 4, 5$). Unlike the unique diagonal alignment required for the best solution, there are combinatorially many ways to form these suboptimal clumps. The normalisation constraint stabilises these high-loss configurations, effectively trapping the dynamics in a high-energy part.

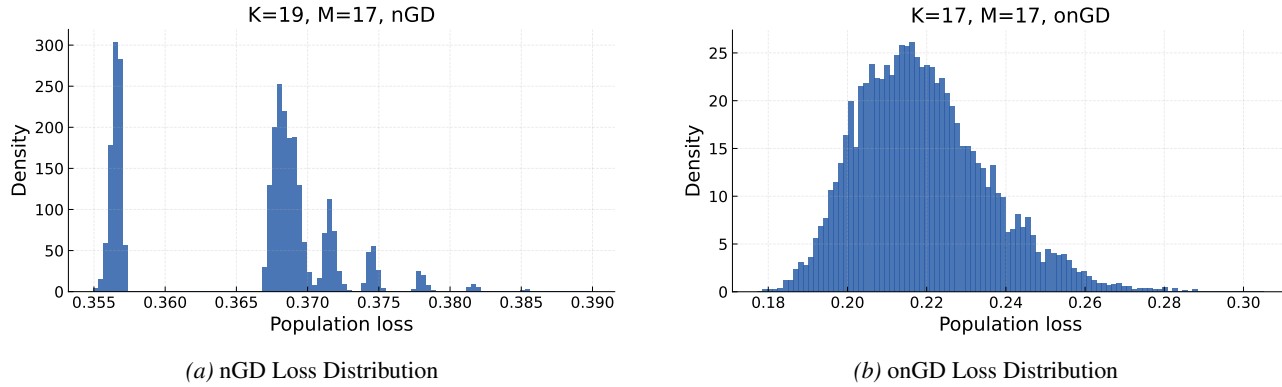

*(a)* nGD Loss Distribution

*(b)* onGD Loss Distribution

*Figure 13.* **Loss distribution in different constraints** These histograms show the empirical density of the final population loss after 20,000,000 running steps. **(a)** Normalized Gradient Descent (nGD) with $K = 19, M = 17$. Despite overparameterisation, the loss distribution is strictly multimodal, exhibiting discrete peaks at high loss values. **(b)** Orthonormalised Gradient Descent (onGD) with $K = 17, M = 17$. The distribution is unimodal and broad, concentrated entirely in a high-error regime (centered at 0.22).

### F.3. Orthonormalized Student Setting (onGD)

In Orthonormalized Gradient Descent, we enforce the stronger constraint that the student weights remain orthonormal throughout training, i.e., $WW^\top = dI_K$ (implying $Q = I_K$). The update rule projects the gradient onto the tangent space of the Stiefel manifold:

$$\frac{dW}{dt} = -\eta\nabla_W\mathcal{L} + \frac{\eta}{d}(\nabla_W\mathcal{L}\,W^\top)W, \quad (57)$$

where $W \in \mathbb{R}^{K\times d}$. Under this setting, the student-student covariance matrix is fixed, so:

$$\frac{dQ_{ik}}{dt} = 0. \quad (58)$$

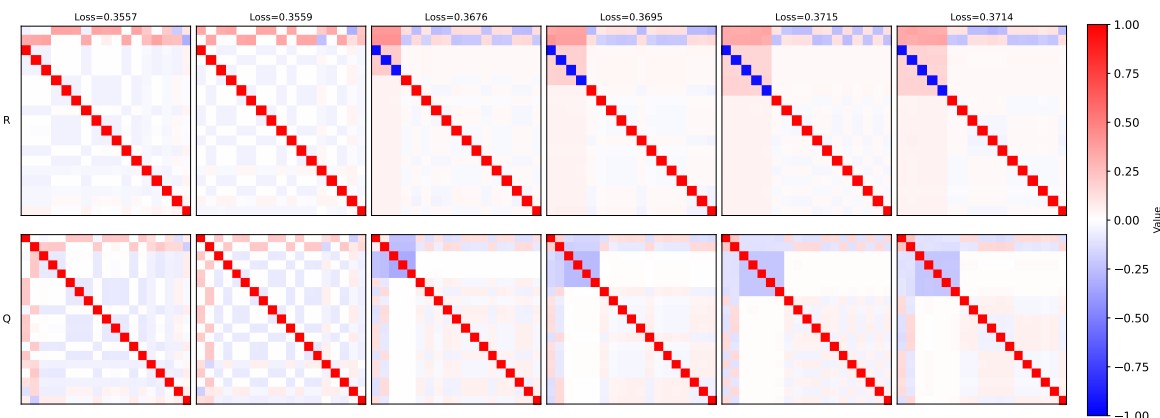

*Figure 14.* **Order parameters of local minima in nGD ($K = 19, M = 17$).** The columns correspond to different final losses. **Left Two Columns (Best Minima):** The $R$ matrices exhibit a near-perfect diagonal structure, indicating successful retrieval of the 17 teacher features. However, the irreducible error persists because the 2 excess students (constrained to $Q_{ii} = 1$) cannot decay to zero. **Right Four Columns (Suboptimal Minima):** These states represent the dominant local minima cluster. They exhibit "blocking" defects (e.g., 2-to-1 assignments seen as red blocks in $R$), where extra units are misallocated, further elevating the loss.

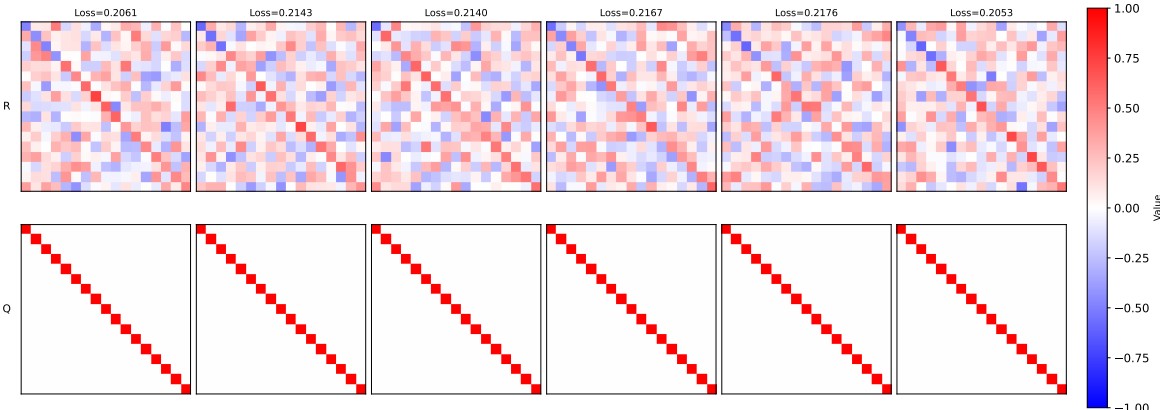

*Figure 15.* **Final order parameters for onGD ($K = M = 17$).** The columns correspond to different final losses. **Bottom Row:** The student-student overlap matrices $Q$ retain a strict identity structure ($Q = I$), satisfying the orthogonality constraint. **Top Row:** The student-teacher overlap matrices $R$ exhibit a disordered, noise-like pattern. Unlike successful learning scenarios shown in Fig. 7, the entries remain diffuse and small. This lack of structural alignment confirms that the student units fail to retrieve the teacher vectors in this constrained setting.

The dynamics of the student-teacher overlap $R$ are modified by a mixing term involving all other student units:

$$\frac{dR_{in}}{dt} = \eta v_i \left[ \sum_{m=1}^{M} v_m^* I_3(i, n, m) - \sum_{j=1}^{K} v_j I_3(i, n, j) \right] - \eta v_i \sum_{k=1}^{K} R_{kn} \left[ \sum_{m=1}^{M} v_m^* I_3(i, k, m) - \sum_{j=1}^{K} v_j I_3(i, k, j) \right]. \quad (59)$$

### F.4. Empirical Observations for Orthonormalized Gradient Descent

We next examine the learning dynamics of Orthonormalized Gradient Descent (onGD). We focus on the student-teacher setting with $K = M = 17$. Fig. 13b presents the distribution of the final population loss.

As shown in Fig. 13b, onGD does not achieve meaningful learning. The final loss distribution is concentrated in the high-error regime of $[0.18, 0.30]$, with a peak around $0.22$. The structural reason for this failure is evident in Fig. 15: the entries of the student-teacher overlap matrix $R$ remain disordered, confirming that the network is unable to retrieve the teacher configuration.

The failure of onGD appears to be closely related to the excessive rigidity imposed by the orthogonality constraint. In a typical successful learning trajectory (as seen in unconstrained GD), student units often pass through intermediate phases

| | | (a) Results for case $K = M$. | | |
|---|---|---|---|---|
| **K** | **M** | **Local (%)** | **Global (%)** | **None (%)** |
| 6 | 6 | 0.91 | 98.82 | 0.27 |
| 7 | 7 | 5.30 | 94.37 | 0.33 |
| 8 | 8 | 13.16 | 86.09 | 0.75 |
| 9 | 9 | 21.15 | 78.53 | 0.32 |
| 10 | 10 | 33.27 | 66.38 | 0.35 |
| 11 | 11 | 43.74 | 55.92 | 0.34 |
| 12 | 12 | 54.62 | 45.12 | 0.26 |
| 13 | 13 | 63.87 | 35.93 | 0.20 |
| 14 | 14 | 71.69 | 28.08 | 0.23 |
| 15 | 15 | 77.99 | 21.81 | 0.20 |
| 16 | 16 | 82.48 | 17.44 | 0.08 |
| 17 | 17 | 86.79 | 13.18 | 0.03 |
| 18 | 18 | 90.42 | 9.51 | 0.07 |
| 19 | 19 | 92.54 | 7.38 | 0.08 |
| 20 | 20 | 94.76 | 5.21 | 0.03 |

| | | (b) Results for the overparameterised case $K = M + 1$. | | |
|---|---|---|---|---|
| **K** | **M** | **Local (%)** | **Global (%)** | **None (%)** |
| 9 | 8 | 0.07 | 99.74 | 0.19 |
| 10 | 9 | 0.75 | 98.95 | 0.30 |
| 11 | 10 | 1.93 | 97.26 | 0.81 |
| 12 | 11 | 3.83 | 94.25 | 1.92 |
| 13 | 12 | 5.74 | 91.42 | 2.84 |
| 14 | 13 | 8.37 | 87.46 | 4.17 |
| 15 | 14 | 13.50 | 81.16 | 5.34 |
| 16 | 15 | 18.12 | 75.36 | 6.52 |
| 17 | 16 | 28.58 | 65.47 | 5.95 |
| 18 | 17 | 36.88 | 57.50 | 5.62 |
| 19 | 18 | 42.25 | 52.62 | 5.13 |
| 20 | 19 | 51.46 | 44.24 | 4.30 |

*Table 3.* **Statistics of minima under mean-field dynamics.** Proportions of local, global, and non-minima found by integrating the mean-field ODEs across 10,000 random seeds for each $(K, M)$ pair.

where they become correlated ($Q_{ik} \neq 0$) to "sense" the teacher's structure. By enforcing $Q_{ik} = 0$ at $i \neq k$, onGD effectively forbids these cooperative intermediate states. Our results suggest that for $K = 17$, the gradient flow on this constrained manifold lacks accessible trajectories from the random initialization to the teacher configuration, effectively locking the system in a high-loss state.

To comparison, we present representative learning trajectories in a setting with few hidden units ($K = M = 2$). As shown in Fig. 16, successful learning is possible in this regime, indicating that the obstruction induced by the orthogonality constraint is dimension-dependent: systems with only a few hidden units can still navigate the Stiefel manifold to find the solution.

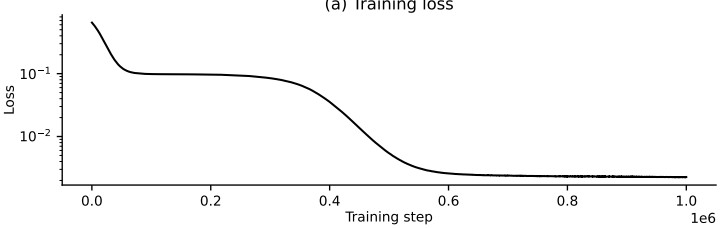
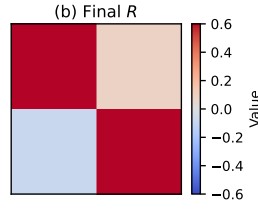

*Figure 16.* **Dynamics of onGD in a setting with small hidden layers** ($K = M = 2$). **(a)** The training loss (log scale) decreases monotonically and converges to a near-zero value ($\sim 10^{-3}$), indicating successful optimisation. **(b)** The final student-teacher overlap matrix $R$ exhibits a clear diagonal structure (red blocks indicate high positive correlation). This confirms that, unlike the case with larger hidden layers ($K = M = 17$), the orthogonality constraint perfectly retrieves the teacher weights in low dimensions.

## G. Spurious minima in two-layer ReLU networks

We reproduce the gradient descent results for ReLU soft committee machines using mean-field dynamics over 10,000 random initialisations. Table 3a reports the proportions of global minima, spurious local minima, and non-minima in the balanced case $K = M$, while Table 3b presents the corresponding statistics in the overparameterised setting $K = M + 1$. The observed proportions are in close agreement with those reported in Safran & Shamir (2018).

## G.1. Extended Empirical Results for Constrained Dynamics

In this section, we provide a more granular look at the empirical convergence properties of the constrained dynamics, specifically Normalized Gradient Descent (nGD) and Orthonormalized Gradient Descent (onGD).

Tables 4 and 5 report the statistical breakdown of the final states reached by nGD for the well-specified ($K = M$) and mildly overparameterised ($K = M + 1$) regimes, respectively. Following the methodology of Safran & Shamir (2018), we define an empirical threshold $\varepsilon$ to classify a run as having reached a global minimum, and a gradient norm threshold $\delta$ (evaluated on $dQ/dt$ and $dR/dt$) to identify non-convergent trajectories.

As reported in the rightmost columns of the tables, these thresholds were selected via visual inspection of the loss histograms and vary depending on the network width. Notably, we observe a general slowdown in the dynamics for the normalised case compared to unconstrained gradient descent. This slowdown is particularly pronounced in lower-dimensional configurations, where the system is more heavily restricted by the constraint. Because the spherical constraint shrinks the feasible space and alters the local geometry, navigating to the global minimum becomes progressively harder, requiring careful tuning of the threshold.

*Table 4.* **Statistics of minima in nGD under mean-field dynamics for case K=M.** Proportions of local, global, and non-minima obtained by integrating the mean-field ODEs under Normalized GD over 10,000 random seeds for each $(K, M)$ with $K = M$. A result is classified as a global minimum when its error is smaller than $\varepsilon$. For the result whose errors are larger than $\varepsilon$, if is gradients ($dQ_{ij}/dt$ or $dR_{in}/dt$) are larger than $\delta$, it will be classified as None. Otherwise it will be classified as a local minimum. From the analysis in Section F, the global minima of nGD are very large.

| K | M | Local (%) | Global (%) | None (%) | $\varepsilon$ | $\delta$ |
|---|---|---|---|---|---|---|
| 2 | 2 | 0.00 | 99.77 | 0.23 | 1e-2 | 1e-2 |
| 3 | 3 | 0.15 | 99.85 | 0.00 | 1e-2 | 1e-2 |
| 4 | 4 | 0.02 | 99.98 | 0.00 | 1e-2 | 1e-2 |
| 5 | 5 | 0.01 | 99.99 | 0.00 | 1e-2 | 1e-2 |
| 6 | 6 | 0.00 | 100.00 | 0.00 | 1e-2 | 1e-2 |
| 7 | 7 | 0.62 | 99.38 | 0.00 | 1e-2 | 1e-2 |
| 8 | 8 | 4.87 | 95.13 | 0.00 | 1e-2 | 1e-2 |
| 9 | 9 | 12.93 | 87.07 | 0.00 | 1e-2 | 1e-2 |
| 10 | 10 | 24.15 | 75.85 | 0.00 | 1e-2 | 1e-2 |
| 11 | 11 | 36.45 | 63.55 | 0.00 | 1e-2 | 1e-2 |
| 12 | 12 | 49.95 | 50.05 | 0.00 | 1e-2 | 1e-2 |
| 13 | 13 | 58.57 | 41.43 | 0.00 | 1e-2 | 1e-2 |
| 14 | 14 | 67.92 | 32.08 | 0.00 | 1e-2 | 1e-2 |
| 15 | 15 | 75.37 | 24.63 | 0.00 | 1e-2 | 1e-2 |
| 16 | 16 | 81.49 | 18.51 | 0.00 | 1e-2 | 1e-2 |
| 17 | 17 | 85.88 | 14.12 | 0.00 | 1e-2 | 1e-2 |
| 18 | 18 | 89.77 | 10.23 | 0.00 | 1e-2 | 1e-2 |
| 19 | 19 | 92.01 | 7.99 | 0.00 | 1e-2 | 1e-2 |
| 20 | 20 | 94.38 | 5.62 | 0.00 | 1e-2 | 1e-2 |

*Table 5.* **Statistics of minima in nGD under mean-field dynamics for case K=M+1.** Proportions of local, global, and non-minima obtained by integrating the mean-field ODEs under Normalized GD over 10,000 random seeds for each $(K, M)$ with $K = M + 1$. A result is classified as a global minimum when its error is smaller than $\varepsilon$. For the result whose errors are larger than $\varepsilon$, if is gradients ($dQ_{ij}/dt$ or $dR_{in}/dt$) are larger than $\delta$, it will be classified as None. Otherwise it will be classified as a local minimum. From the analysis in Section F, the global minima of nGD are very large.

| K | M | Local (%) | Global (%) | None (%) | $\varepsilon$ | $\delta$ |
|---|---|---|---|---|---|---|
| 3 | 2 | 2.73 | 97.27 | 0.00 | 0.123 | 1e-1 |
| 4 | 3 | 48.95 | 51.05 | 0.00 | 0.11 | 1e-1 |

Continued on next page

| K | M | Local (%) | Global (%) | None (%) | $\varepsilon$ | $\delta$ |
|---|---|-----------|------------|----------|---------------|----------|
| 5 | 4 | 1.34 | 98.66 | 0.00 | 0.105 | 1e-1 |
| 6 | 5 | 1.40 | 98.60 | 0.00 | 0.105 | 1e-1 |
| 7 | 6 | 2.87 | 97.12 | 0.01 | 1e-1 | 1e-1 |
| 8 | 7 | 1.80 | 98.19 | 0.01 | 1e-1 | 1e-1 |
| 9 | 8 | 1.49 | 98.48 | 0.03 | 1e-1 | 1e-1 |
| 10 | 9 | 2.57 | 97.42 | 0.01 | 1e-1 | 1e-1 |
| 11 | 10 | 4.25 | 95.74 | 0.01 | 1e-1 | 1e-1 |
| 12 | 11 | 6.61 | 93.39 | 0.00 | 1e-1 | 1e-1 |
| 13 | 12 | 10.58 | 89.42 | 0.00 | 1e-1 | 1e-1 |
| 14 | 13 | 15.64 | 84.36 | 0.00 | 1e-1 | 1e-1 |
| 15 | 14 | 21.55 | 78.45 | 0.00 | 1e-1 | 1e-1 |
| 16 | 15 | 27.54 | 72.46 | 0.00 | 1e-1 | 1e-1 |
| 17 | 16 | 34.73 | 65.27 | 0.00 | 1e-1 | 1e-1 |
| 18 | 17 | 41.66 | 58.34 | 0.00 | 1e-1 | 1e-1 |
| 19 | 18 | 48.76 | 51.24 | 0.00 | 1e-1 | 1e-1 |
| 20 | 19 | 55.83 | 44.17 | 0.00 | 1e-1 | 1e-1 |

In contrast to nGD, we observe a complete failure of convergence when enforcing the stricter Stiefel manifold constraints of onGD. Because the network never successfully reaches the global minimum in our simulations, tabulating the convergence statistics gives no additional insights. Instead, we visualise the full empirical distribution of the final population loss in Figures 17 and 18.

The top rows of both figures display the well-specified case ($K = M$), while the bottom rows show the overparameterised case ($K = M + 1$). Notice that the final loss distributions for onGD do not exhibit the discrete, quantised band structure characteristic of both unconstrained GD and nGD. This lack of quantisation is a direct consequence of the orthogonality constraint. Indeed, the compensation mechanism—which relies on aligned neurons adjusting their partial overlaps and magnitudes to offset the error of anti-aligned units—is strictly forbidden on the Stiefel manifold. Without the ability to form these cooperative partial overlaps, the network cannot settle into the structured spurious families.

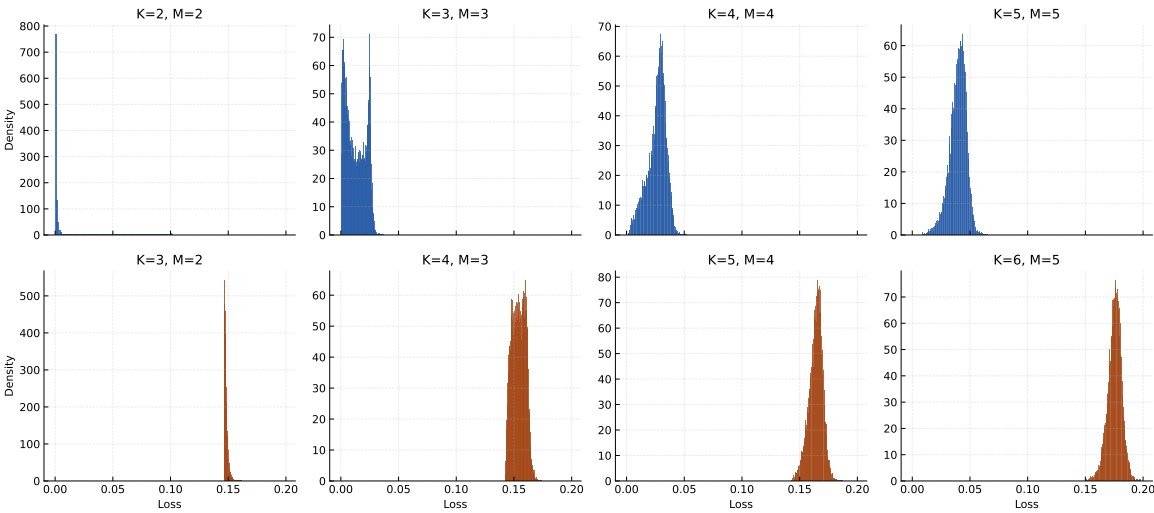

*Figure 17.* **Distribution of loss for small networks in onGD under mean-field dynamic.** Histograms of the final population loss obtained after long-time integration of the mean-field ODEs under Orthonormalised GD over 10,000 random initialisations. Each panel corresponds to a small student-teacher size configuration $(K, M)$. The top row shows the equal case $K = M$ and the bottom row shows the overparameterised case $K = M + 1$. All panels share the same horizontal axis for direct comparison.

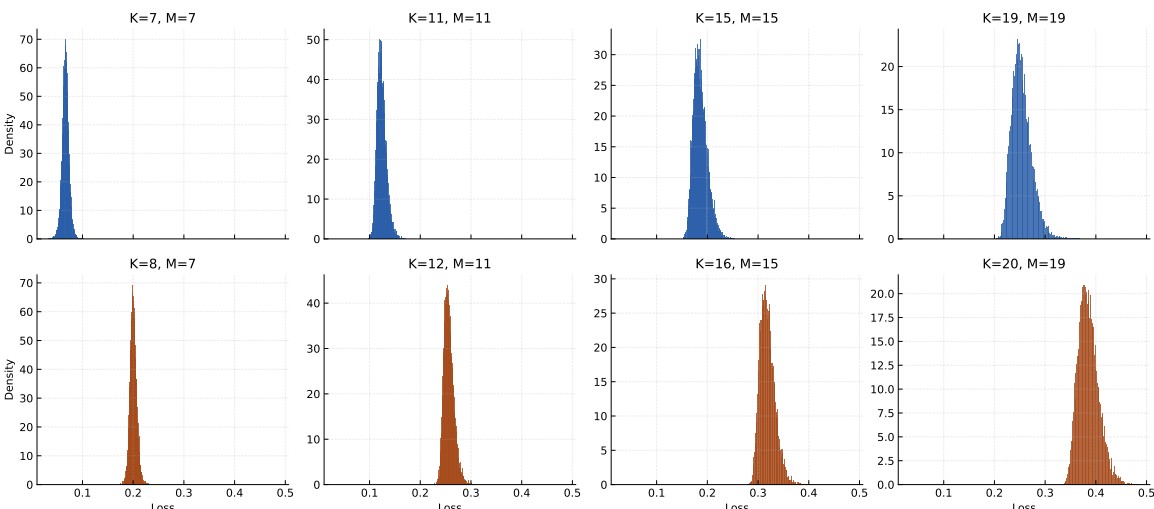

*Figure 18.* **Distribution of loss for wider networks in onGD under mean-field dynamic.** Histograms of the final population loss obtained after long-time integration of the mean-field ODEs under Orthonormalised GD over 10,000 random initialisations. Each panel corresponds to a student-teacher size configuration $(K, M)$ with larger widths than those shown in Fig. 17. The top row shows the equal case $K = M$ and the bottom row shows the overparameterised case $K = M + 1$. All panels share the same horizontal axis for direct comparison.

## H. Numerical and Computational Details

The large-scale simulations reported in this work were run on a CPU cluster. The CPU nodes used in our experiments are equipped with two Intel Xeon Gold 6130 processors, providing 32 CPU cores per node and 96 GiB of RAM per standard node. No GPU acceleration was used.

For the main ODE simulations, loss histograms, alternative-activation experiments, and constrained-dynamics experiments, we used Slurm job arrays to run independent random initialisations in parallel. For settings with both $K < 15$ and $M < 15$, each trajectory was run with one CPU core. For larger settings, where at least one of $K$ or $M$ is at least 15, each trajectory was run with 2 CPU cores. Each array job was assigned a wall-clock limit of 24 hours. The largest runs required up to $10^7$ training steps in the well-specified case $K = M$, and up to $1.2 \times 10^7$ training steps in the overparameterised case $K = M + 1$.

The stability analysis experiments are substantially smaller and can be reproduced on a standard workstation. For example, on a laptop/desktop CPU comparable to an Intel i7-13650HX with 32 GB RAM, the stability analysis experiments reported in Appendix D complete within approximately two hours. These experiments do not require GPU resources.

Memory usage is modest for all experiments because the simulations evolve the order parameters $Q$, $R$ and $T$, rather than full high-dimensional network weights, except in the finite-dimensional robustness checks. The dominant cost is therefore wall-clock time over many independent initialisations rather than memory.

