# OpenReview forum: "Sharp description of local minima in the loss landscape of high-dimensional two-layer ReLU neural networks"
_ICML.cc/2026/Conference — ICML 2026 regular_

### Official Review · Reviewer_WvpR · 2026-03-04

**Soundness:** 3
**Presentation:** 2
**Significance:** 2
**Originality:** 2
**Overall Recommendation:** 2
**Confidence:** 3

**Summary:**

This paper studies how the loss landscape of two-layer ReLU networks looks and how it affects training. Using a teacher-student setup with Gaussian inputs, the authors show that when the student network has the same width as the teacher, the landscape has lots of isolated “bad” minima where gradient descent can get stuck, leading to discrete loss levels. But if you add even one extra neuron, the landscape changes dramatically: some bad minima disappear and the remaining ones merge into connected, flat regions, letting the optimizer move freely and reach the global minimum much more easily. They also show that these patterns explain why gradient flow and SGD tend to behave the way they do.

**Compliance With Llm Reviewing Policy:**

Affirmed.

**Final Justification:**

I am still unconvinced about the broader value or impact of the results as currently presented. A clearer positioning relative to existing literature would help in appreciating the significance. In its current state, I stay with my position.

**Key Questions For Authors:**

1) Your analysis focuses on two-layer ReLU networks with fixed second layers. do you expect your results about the connectivity of minima and over-parameterisation effects to extend to deeper networks or networks with trainable second layers? Do you expect this to hold in the noise multi index setting?

2) Recent works on single/multi index settings, such as [1], [2] and more, show that SGD can align weights with principal subspaces in more general settings. How do you see your results complementing or differing from these broader structural convergence results? This is actually my main concern, since it seems some of the key questions may have already been addressed, though I could be convinced otherwise and would happily increase my score if I’ve misunderstood.

**Limitations:**

yes

**Strengths And Weaknesses:**

$\textbf{Strengths}$

This work gives an analysis of two-layer ReLU networks in the teacher-student framework, providing characterizations of local minima, their connectivity, and the effect of over-parameterisation on them. Using order parameters and ODEs for gradient flow, the authors explain why having extra neurons helps the network avoid "bad" local minima and lets isolated solutions merge into continuous, flat paths. In short, it gives a clear and intuitive explanation for why gradient-based methods work so well in wide, overparameterised networks.

$\textbf{Weaknesses}$

The analysis in this work is limited to two-layer ReLU networks with a fixed second layer, Gaussian inputs, and orthonormal teacher weights, which restricts its applicability to more complex architectures or real-world data. Several assumptions, such as uniform readout layers, rarely hold in practice. Moreover, the study focuses on idealised, noiseless teacher-student setups, leaving open questions about generalisation and optimisation dynamics in more realistic, noisy, or deeper networks. In contrast, other works tackle similar questions under broader settings: for example, [1] shows that SGD-trained two-layer networks with noisy multiple-index targets converge to a low-dimensional subspace and [2] extend this type of structural result to arbitrary neural networks. Although this paper provides a nice insight for a specific teacher-student setup, its assumptions limit the generality of its conclusions compared to these broader theoretical frameworks.

[1] Mousavi-Hosseini A, Park S, Girotti M, Mitliagkas I, Erdogdu MA. Neural networks efficiently learn low-dimensional representations with SGD. arXiv preprint arXiv:2209.14863. 2022 Sep 29.

[2] Tsikouras N, Pantis Y, Mitliagkas I, Tzamos C. A Derandomization Framework for Structure Discovery: Applications in Neural Networks and Beyond. arXiv preprint arXiv:2510.19382. 2025 Oct 22.

---

> ### Author Rebuttal · Authors · 2026-03-27
>
> We thank the reviewer for their comments and time spent reviewing our work:
>
> *   **Limited framework.** We acknowledge that our Gaussian teacher-student model is an idealised abstraction, but it serves as the standard theoretical baseline for isolating landscape phenomena, as used in previous works and still presents several surprising features. Furthermore, we explicitly address the highlighted constraints in the text: (1) Trainable second layer: We do not rely solely on a fixed second layer. We introduce "Two-Layer GD" (2L-GD) in Section 3.2, where both student weights and readout coefficients are trained simultaneously. Full derivations are provided in Appendix B, and Table 2 confirms the convergence behaviour remains consistent. (2) Orthonormal Teachers: We clarify in Section 2 that this assumption is made strictly for analytical clarity to get simplified expressions for the ansatz. The framework extends straightforwardly to arbitrary teacher configurations. The ODE results in Figs. 1, 2, and 6 actually use standard teacher initialisations. We realise we were not clear on this point and will clarify it. (3) Universality: We explicitly show that these landscape features are not universal to all algorithms. In Section 3.2 and Appendix F.3, we demonstrate that optimising on the Stiefel manifold (Orthonormalized GD) leads to a completely different landscape, as the orthonormality constraint forbids the magnitude adjustments required for the compensation mechanism. To show the structure persists with correlated teachers, we are attaching novel results: across simulations for d=196,392,784, orthonormal (infinity).  \[link https://limewire.com/d/w6OXs#g7gWhw6Vsf please notice that this anonymous link will expire in 7 days\] \[ link to pngs https://anonymous.4open.science/r/Plots_2026-680E \].
>
> *   **"Recent works... show that SGD can align weights with principal subspaces in more general settings. How do you see your results complementing or differing..."** We will certainly include \[1\] and \[2\] in our related work, but they address a fundamentally different aspect of the problem. Most of the SGD results for multi-index models, such as the staircase literature, focus on settings in which the optimisation landscape is non-convex but benign, with only saddle-points and (possibly degenerated) global minima. In this context, the relevant question is establishing the convergence rate (or equivalently for one-pass SGD, sample complexity) at which the weights align with each relevant subspace. Our work instead investigates a setting where the population landscape can have spurious local minima, which SGD will not escape. Results for which the population landscape is highly non-convex are considerably scarcer in the literature, and characterizing them is a challenging problem — hence why the “simple” model.
>
> *   Despite being fundamentally different, our results can also be seen as complementary to this literature on a higher-level picture. Indeed, one can imagine a more realistic problem as having a combination of both aspects: subspaces which are connected by saddle points, and therefore are learned in a saddle-to-saddle fashion, but which eventually can lead to a spurious local minimum. For such a problem, our analysis would help identify the discrete geometrical structure of the local minima, while an analysis such as in \[1\] would quantify the convergence rate/sample complexity required to converge to such a subspace.
>
> *   **"...extend to deeper networks... hold in the noise multi index setting?"** Because the core compensation mechanism and symmetry-breaking patterns we identify are intrinsic to the ReLU non-linearity and the permutation symmetry of hidden units, we strongly expect the qualitative hierarchy of minima to persist in noisy multi-index settings, even if the precise quantized loss levels shift. Extending this exact analytical mean-field description to deep networks is mathematically challenging, but broader empirical evidence in the literature suggests the geometric intuition holds: overparameterization continues to turn isolated minima into connected, flat manifolds. (Note: As mentioned above, trainable second layers are already addressed via 2L-GD).

---

> > ### Author Rebuttal · Reviewer_WvpR · 2026-04-04
> >
> > Thank you for the answers to my questions.
> >
> > I have a follow-up regarding the distinction between population landscapes and finite-sample analysis. I understand that your work provides valuable insights into spurious local minima and connectivity at the population level, which can be seen as a “best-case” or idealized scenario. However, in practice, training is always done with finite samples and noisy optimization methods like SGD. From this perspective, one might argue that finite-sample analysis, including generalization bounds and convergence rates, could be more directly informative about what can actually be achieved. I am interested in your view on how the insights from the population landscape could translate to the finite-sample regime, and whether you see population-level results as mostly providing lower bounds on what is possible in practice.

---

> > > ### Author Response · Authors · 2026-04-07
> > >
> > > We thank the reviewer for the follow-up question. On the relevance of understanding the population vs. empirical landscape, it depends on the practical task.
> > >
> > > Indeed, as pointed out by the reviewer, in practice one uses a noisy stochastic algorithm for optimisation, such as a variation of SGD, and the data is split in independent batches which are successively used across different epochs. The classical result by Monro and Robbins (1951) shows that during the first epoch, SGD is on average going down the population landscape, also known as **one-pass** regime. As data is re-visited in the successive epochs, the gradient becomes biased, and SGD transitions to a **multi-pass** regime. While the one-pass regime is always relevant for the first epoch, whether this is enough to understand its convergence is task dependent. For instance, when training LLMs on billions of token it is fairly common to run for less than an epoch (see caption of Table 2.2 in [1] for GPT3). On the other hand, when training a CNN on ImageNet, hundreds of epochs are needed.
> > >
> > > Although we focus most of the discussion in the main manuscript on gradient flow, our results are readily generalised to one-pass SGD. In Appendix B.6, we derive the evolution of the summary statistics under SGD (Eqs. 41–43). When compared to the population-level gradient flow (Eqs. 36–38), the difference appears as a term that scales with $\eta^2$. This suggests that, for small enough learning rates, SGD largely follows the same geometric structure defined at the population level, with deviations that can be understood as higher-order corrections. We also checked this empirically. In additional finite-sample SGD experiments (to be included in the camera-ready version, see tables below), the proportion of initialisations that reach the global minimum versus spurious local minima closely matches what the population-level theory predicts, even in the presence of noise.
> > >
> > > On the theoretical side most analyses of one-pass SGD in the literature have focused on either convex landscapes or landscapes that contain only global minima and strict saddles (i.e. saddles that have a direction pointing to a global minimum). These preclude the study of important phenomena, such as how the landscape geometry depends on overparametrisation. On the other hand, mathematically tractable settings where the population landscape has local minima are much scarcer. Our work is part of this endeavour.
> > >
> > > | Kind              | K    | M    | Local (%) | Global (%) | None (%) | Seed Num |
> > > | ----------------- | ---- | ---- | --------- | ---------- | -------- | -------- |
> > > | Finite-sample SGD | 19   | 19   | 94        | 6          | 0        | 100      |
> > > | Finite-sample SGD | 20   | 19   | 30        | 70         | 0        | 100      |
> > > | Population GD     | 19   | 19   | 92.54     | 7.38       | 0.08     | 10,000   |
> > > | Population GD     | 20   | 19   | 51.46     | 44.24      | 4.30     | 10,000   |
> > >
> > > *The table is obtained using a dataset of 500,000 samples and training for 10,000,000 steps. Notice that ReLU doesn't have a known solution for the term $I_4$ in the ODEs, therefore we rely on numerical simulations, which are more computationally demanding and we consider 100 different initialisations. The result is compared with the population GD already reported in the paper.*
> > >
> > > [1] Language Models are Few-Shot Learners, https://arxiv.org/pdf/2005.14165

---

### Official Review · Reviewer_7QNk · 2026-03-09

**Soundness:** 2
**Presentation:** 2
**Significance:** 2
**Originality:** 3
**Overall Recommendation:** 4
**Confidence:** 3

**Summary:**

The paper studies the loss landscape of two-layer ReLU networks on Gaussian teacher-student data. By analyzing the training dynamics under gradient flow, it characterizes minima in terms of some summary statistics. With empirical simulations using this description, the paper identifies a hierarchy among minima and provides a geometric interpretation of it. The authors show that mild overparametrization influences the separation and stability of these minima. All of this is extended to more activations and algorithms.

**Compliance With Llm Reviewing Policy:**

Affirmed.

**Final Justification:**

After discussion with the authors, I decided to update my final score from 3 to 4. I believe that, conditioned on substantial improvement in positioning relative to other work, the paper will be of value to the research community.

**Key Questions For Authors:**

I use this space to remark the points in my review where your answers will likely influence my decision the most:
1. The last weakness under “soundness” regarding appendix A.
2. The weakness under “significance” regarding related work.

**Limitations:**

Yes (the authors adequately discussed the limitations and potential negative societal impact of their work).

**Strengths And Weaknesses:**

_**Soundness**_

**Strengths**

1. The formal derivations appear correct.
2. The experimental design is appropriate and sufficiently varied.


**Weaknesses**

3. Introduction: The introduction suggests insight into population error without clarifying that the population loss is optimized directly, which is misleading in my opinion. I think this distinction should be made early.
4. Experiments: Figure 3 studies only two minima, which in my opinion is too weak to “confirm” anything (as claimed in line 300). I would rephrase this.
5. Experiments: In the overparameterized part of Figure 3, I do not see why it is valid to simply append a zero row.
6. Appendix A seems to rely on an unstated assumption/claim that $\frac{d\mathcal{L}}{dt} = 0$ implies $\frac{dQ}{dt}, \frac{dR}{dt} = 0$. I do not see why this holds.

_**Presentation**_

**Strengths**

1. The introduction is very clear for the most part and well-structured.
2. The explanation of the string method in the appendix is very easy to follow.


**Weaknesses**

3. Introduction: it is not specified which contributions are theoretical and which are empirical.
4. Introduction: In the first two contributions, I think it is more fitting to convey the “essence” of the summary statistics and hierarchies rather than just say they exist.
5. Related work: by a quick search into literature subsequent to Safran & Shamir, I found papers you do not mention [1,2] which consider a similar setting and seem highly relevant. The former characterizes minima (and mentions a matrix structure resembling your ansatz) and the latter discusses the effect of overparametrization.

[1] Arjevani and Field, https://arxiv.org/abs/2107.10370

[2] Safran et al., https://arxiv.org/abs/2006.01005

6. Result 1 statement: the process of understanding the exact results is cumbersome: the reader needs to go to Appendix A, which refers to notation introduced in Eq. 10, which actually isn’t quite the required notation
7. Result 2 statement: I think it is more appropriate to state the nature of the distinct families rather than discuss what they allow to analyse.
8. Experiments: in the beginning of section 3 you state that you use the derived dynamics of $Q(t), R(t)$ to perform the experiments. I think it would benefit the reader if you explain how they are used and when it is complemented by other computations.
9. Experiments: in line 220 you describe a compensation mechanism. It seems that the evidence supporting this is in figures 8/9 in the appendix, but I did not see a reference to them.
10. Experiments: In section 3.2, I do not see the value in discussing the Gaussian integral.
11. Appendices: some suggestions: (a) Consider moving appendix B before A, as it derives and states equations which are referred to from A. (b) Consider making your notation more similar to the main body.

_**Significance**_

**Strengths**

1. The paper addresses relevant topics.
2. The scope of the paper is broad in terms of training dynamics.


**Weaknesses**

3. Continuing my comment on related work from the “Presentation” section, I am uncertain of the degree to which the paper offers new insights/understandings.

_**Originality**_

**Strengths**

1. I do not know of other literature exploring the same questions in an empirical manner.
2. In particular, the perspective of the proposed summary statistics and performing experiments based on the them is original.


Also, a small typo: Equation (28) misses $v_i$.

---

> ### Author Rebuttal · Authors · 2026-03-27
>
> We thank the reviewer for their thorough and constructive feedback.
> - **Soundness 3**:  We appreciate the pointer. We will clarify in the early sections of the revised paper that the population loss is optimised directly, as is standard in related works on the topic.
> - **Soundness 4**: We recognise that a single path between two symmetric solutions may appear anecdotal. In our study, we investigated all permutations of the identified minima families. Figure 3 was chosen as a representative visualisation of the general phenomenon: in the well-specified case, symmetry-equivalent solutions are isolated by high-loss barriers, whereas overparameterisation introduces flat directions that connect them. We will rephrase "confirm" to "illustrate" and note that this behaviour is consistent across all observed permutations.
> - **Soundness 5**: The validity of this embedding stems from the fact that a neuron with zero weights is a stationary point of the gradient flow in this architecture; this has been proved in Fukumizu & Amari 2000. By appending a zero row, we demonstrate that the extra degree of freedom allows the system to interpolate between different configurations without incurring a loss penalty.
> - **Soundness 6**: We will elaborate more on this point in a rivised Appendix A: for Gaussian inputs, any stationary point of the high-dimensional population risk ($\nabla_{\mathbf{w}} \mathcal{L} = 0$) is necessarily a stationary point of the summary statistics equations ($\mathcal{F}(Q,R) = 0$). Since the gradient depends on the weights only through their overlaps $Q$ and $R$, no information regarding the stationary condition is lost in this reduced representation.
> - **Presentation 3**: All our primary contributions are theoretical, which we then rigorously verify numerically. We will explicitly state this distinction in the revised introduction.
> - **Presentation 4**: We will follow the recommendation to convey the "essence" of our findings: summary statistics allow us to map high-dimensional weight descriptions to a simplified space without losing information, revealing a highly structured nature in what appears to be a complex problem.
> - **Presentation 6**: We agree that forcing the reader to bounce back and forth to the appendix breaks the flow. We will rewrite the statement of Result 1 in the main text to make it more self-contained and fix the notation referencing Eq. 10 so it strictly aligns with the required format.
> - **Presentation 7**: We will update the Result 2 statement to explicitly describe the nature of these families (e.g., the specific symmetric arrangements and the count of anti-aligned units) rather than just stating they make analysis tractable.
> Presentation 8: We will clarify this at the start of Section 3. Specifically, we will explain that we use numerical integration of the derived ODEs for Q and R to simulate the exact mean-field trajectories, which we then use to complement and validate the finite-width network simulations.
> - **Presentation 9**: We will explicitly reference Figures 8 and 9 around line 220 to point the reader directly to the visual evidence of the compensation mechanism.
> - **Presentation 10**: We included this to show the framework's extensibility beyond ReLU. However, we will condense this paragraph and move the bulk of the integral discussion to the appendix to maintain the flow.
> - **Presentation 11**: We will swap Appendices A and B so the derivation of dynamics precedes the stationary equations, and we will unify the notation to match the main body.
> - **Significance 3**: While previous works like Safran and Shamir and Safran et al. established empirically that spurious minima exist and that mild overparameterization helps avoid them, they did not systematically characterise the geometry of these critical points and a mechanistic understanding of how overparameterization helps was not discussed. Our paper offers two distinct new mathematical and geometric insights: (1) We don't just show that overparameterization works empirically; we map the exact topological phase transition. Using the string method in the order-parameter space, we show that the isolated point-like minima of the well-specified regime literally dissolve and merge into continuous, connected manifolds linked by flat paths. (2) While Arjevani & Field identified related matrix structures, our mean-field approach reduces the landscape to an exact, dimension-independent set of equations. By plugging our block-symmetric ansatz into these equations, we can analytically calculate the exact quantised loss values of these trapped states, perfectly predicting the empirical loss distribution without ever needing to simulate the network dynamics. We will make sure this distinction is stated explicitly in the revised introduction.
>
> **Equation 28 Typos**: We have corrected them, thank you.

---

> > ### Author Rebuttal · Reviewer_7QNk · 2026-04-01
> >
> > Thank you for your detailed response.
> >
> > Firstly, all of my concerns regarding soundness were fully addressed.
> >
> > Regarding presentation, given the willingness to amend the points I raised, my concerns are addressed as well.
> >
> > Regarding significance, aside from your response to me, I have read the other reviews (and responses) as well as the newly-addressed related work. I recognize the theoretical significance of your exact set of equations for the minima, and I recognize that theory sometimes raises new ideas to test empirically. However, I do not see the benefits of simulating the optimization dynamics through these equations over doing so via direct training.
> >
> > The above leads me to the following follow-up question: of your experimental findings (meaning what they demonstrate, such as the effect of over-parameterization on loss "bins"), which were not previously shown empirically, or proven theoretically by related work mentioned in the reviews?
> >
> > Thank you for your time.

---

> > > ### Author Response · Authors · 2026-04-01
> > >
> > > 1. **Why use macroscopic equations instead of direct training?** Direct training is noisy and high-dimensional (hence prompt to finite-size effects), so at best it gives an empirical picture of a local minimum. In contrast, our summary-statistics equations let us strip away that noise and focus on the underlying optimisation dynamics. This gives us two advantages: first, we can study the dynamics without finite-size artifacts; second, we can go beyond simulation and solve for the fixed points analytically ($\mathcal F = 0$), which gives the exact theoretical loss values at the minima.
> > >
> > > 2. **What is genuinely new compared to prior work?** Earlier work established the basics of spurious minima, but our framework leads to several new insights. Previous studies, like Safran et al. (2018, 2021) and Arjevani & Field (2020), mostly relied on local Hessian analysis, showing that overparameterisation can turn spurious minima into saddle points by introducing negative curvature. What was missing is the global picture. Using the string method, we show that the energy barrier disappears entirely. This reveals flat, continuous manifolds that connect solutions that were previously thought to be isolated. We also find that the loss exhibits a quantised structure, which was not observed in previous works. With a block-symmetric ansatz, we can both explain why this happens and compute the exact loss values of these discrete "bins." While a block-like structure also emerged in Arjevani & Field (2020), their result is different and comes from analysing the local Hessian, which gives less insight into generalisation. The underlying reason for the structure is "symmetry" in both cases; **however**, our approach gives more insight into the consequences for the loss. In particular, it allowed us to identify the compensation mechanism that gives rise to this structure as well as analytically predict the exact loss values of the different bins.
> > >
> > > In short, earlier work focused on local curvature or empirical observations, whereas our approach provides a global, continuous, and fully analytic view of the loss landscape.

---

### Official Review · Reviewer_Xuot · 2026-03-09

**Soundness:** 1
**Presentation:** 3
**Significance:** 2
**Originality:** 2
**Overall Recommendation:** 2
**Confidence:** 3

**Summary:**

**Problem.** Characterising the geometry of the population loss landscape of two-layer ReLU networks $\phi(x,W) = \sum_{k=1}^{K} \text{ReLU}(w_k^\top x / \sqrt{d})$ in a teacher–student setting with Gaussian inputs, as a function of the parameterisation regime (well-specified $K = M$ vs. overparameterised $K > M$).

**Approach.** The paper uses the mean-field formalism of summary statistics (order parameters $Q_{ij} = w_i^\top w_j / d$, $R_{im} = w_i^\top w_m^* / d$) to reduce the high-dimensional landscape to a low-dimensional system of self-consistent equations. Connectivity between minima is probed via the zero-temperature string method in order-parameter space with an induced metric.

**Main results.**
- (R1) Exact, dimension-independent self-consistent equations for stationary points of the population risk (Result 1, §2).
- (R2) Classification of local minima into discrete families indexed by $k_1 \in [0, M]$, with analytical loss values that sharply match the empirical histogram peaks (Result 2, §3.1, Figs 1–2).
- (R3) A topological phase transition: in the well-specified regime, minima are isolated; overparameterisation ($K > M$) creates flat directions that merge them into connected manifolds, destabilising low-order spurious states (§3.1, Figs 3–5).
- (R4) Landscape equivalence for SGD in the appropriate scaling limit (Result 3, §3.2).

**Compliance With Llm Reviewing Policy:**

Affirmed.

**Key Questions For Authors:**

1. **Q1. Completeness of the ansatz.** The block-symmetric ansatz (Eq 8) is the central structural assumption. Can you provide theoretical evidence that *all* local minima conform to this structure? For instance, can you show that any stationary point of the self-consistent equations (Result 1) must have a permutation symmetry compatible with the ansatz, at least in the $K = M$ case?


2. **Q2 Beyond orthonormal teachers.** You assume $T = I_M$ (orthonormal teacher weights). How sensitive are the families to correlated teachers? In particular, does the $k_1$-indexed structure persist when teacher weight vectors have non-trivial overlaps?

**Limitations:**

It is very unclear to be how the paper discussed the limitations of their analysis. Every thing seems completely informal, and at this stage, I do not know if we can accept such papers in ML conference journals.

**Strengths And Weaknesses:**

### Strenghts :
- Nice self-concordant equations in Result one, and story line seems nice and neat. The string method analysis (Figs 3, 5) provides experimental evidence for the topological transition.


### Weaknesses :
- The main paper is very weirdly written, non of the result are put under a mathematical form, and even if every idea seems well exposed, it is hard to understand what result is informal, proven or simply experimental.

---

> ### Author Rebuttal · Authors · 2026-03-27
>
> We thank the reviewer for their comments and time spent in reviewing our work.
> *   **"The main paper is very weirdly written..."** We used a physics-style exposition, focusing on an intuitive and constructive description of the results while reporting the technical aspects in the appendix. We understand this made it hard to tell what is formally proven versus purely empirical. We are happy to restructure the main text into a maths-style theorem-based format to make the claims clearer.
> *   **Q1. Completeness of the ansatz.** No, our results are a combination of analysing the mean-field equations and observing the data. We are not able to formally prove that our ansatz is the _only_ solution. However, we show that it has strong predictive power: it perfectly explains the scaffolding of the fixed points into quantised groups, and it allows us to identify the exact loss values of these minima analytically, without even running the dynamics. This is backed up by extensive numerical verification.
> *   **Q2. Beyond orthonormal teachers.** We clarify in Section 2 that the orthonormal teacher assumption is made strictly for analytical clarity to get simplified expressions for the ansatz. The framework itself extends straightforwardly to arbitrary teacher configurations. In fact, the ODE results shown in Figs. 1, 2, and 6 actually use standard normal teacher initialisations, not orthonormal ones. We realise we were not clear on this point and will clarify it in the text.
> * **Q2. Does the structure persist with correlated teachers?** Yes, it does. To show this, we are attaching some novel results: we ran simulations under different initialisations for d=196,392,784, orthonormal (infinity). \[link to pdf https://limewire.com/d/w6OXs#g7gWhw6Vsf please notice that this anonymous link will expire in 7 days\] \[ link to pngs https://anonymous.4open.science/r/Plots_2026-680E \]. Across a selection of 10 different simulations, the order parameter R consistently follows the exact block-symmetric structure we identified. While a formal IFF (if and only if) proof may not be mathematically possible here, these new tests are strong evidence that the system converges to solutions structured as in our proposed ansatz.

---

> > ### Author Rebuttal · Reviewer_Xuot · 2026-04-01
> >
> > I stand with my position, the results of the paper are a bit to weak for acceptance.

---

### Official Review · Reviewer_emT6 · 2026-03-12

**Soundness:** 3
**Presentation:** 2
**Significance:** 2
**Originality:** 2
**Overall Recommendation:** 4
**Confidence:** 2

**Summary:**

The paper studies the landscape and the minima geometry for data following a summation of M ReLUs, learned by gradient flow with a similar ReLU network with K neurons. The paper specifically focuses on the cases where $M=K$ and where $K>M$, i.e, under over-parameterization. Previous works had already established the existence of spurious local minima for this exact data model, and the fact that over-parameterization helps reduce them. This paper states that for $M=K$, spurious local minima exist and are isolated with quantized values.  Furthermore, even slightly increasing $K$, significantly expands the basin of attraction of the global minimum, with nearly all trajectories converging to zero loss, and the paper further shows that almost all minima in this cases are connected.

**Compliance With Llm Reviewing Policy:**

Affirmed.

**Final Justification:**

I remain borderline about this work. I find the approach and some of the main results interesting, however the presentation could be significantly improved (as mentioned by Reviewer Xuot) by a major revision.

**Key Questions For Authors:**

Please see the last section.

**Limitations:**

yes

**Strengths And Weaknesses:**

Strengths:
- The results are sound and several experiments corroborate the main claims of the paper. The claim that over-parameterization helps with spurious local minima is intuitive and this paper helps to make this claim more rigorous.

Weaknesses:
- The studied problem has been considered in few previous works, however it remains unclear how the present work differs from them and what the advantages of current work are compared to them. The discussion on theoretical results, how they are derived and their novelty are largely missing from the paper. More importantly, given the complex nature of the equations, it remains unclear how useful these derivations can be.

- The paper benefits from a discussion on previous works, specifically  Safran and Shamir 2018, as well as other closely related papers which are not cited [1,2].

- The studied model is too specific to be considered useful. The fact that both the teacher and student have same activation and mathematical form as well as the specific form they take, restricts the significance and applicability of results.


[1]-The Effects of Mild Over-parameterization on the Optimization Landscape of Shallow ReLU Neural Networks. Sarfan et al 2021.

[2]-Annihilation of Spurious Minima in Two-Layer ReLU Networks. Arjevani and Field 2022.

---

> ### Author Rebuttal · Authors · 2026-03-27
>
> We thank the reviewer for their comments and time
> *   **"The discussion on theoretical results, how they are derived and their novelty are largely missing from the paper."** As the mean-field technique itself is not novel, we placed the step-by-step derivation in Appendix B. The originality of our work stems from leveraging this mean-field approach to analytically study the fixed points of the dynamics and characterise the landscape.
> *   **"More importantly, given the complex nature of the equations, it remains unclear how useful these derivations can be."** While the expanded scalar equations presented in Appendix A.2 do appear lengthy, the core mathematical formulation is actually not complex. As shown in Result 1 and detailed in Appendix B.3, the governing dynamics are captured by a compact matrix representation. The apparent complexity only arises when we insert our block-symmetric ansatz into these exact matrix equations and unfold the terms. The fact that this highly constrained, low-dimensional ansatz allows us to study the landscape of complex, high-dimensional neural networks is very interesting. It underlines that the inherently non-convex optimisation landscape has a simple geometric structure.
> *   **"The paper benefits from a discussion on previous works, specifically Safran and Shamir 2018, and related papers which are not cited \[1,2\]."** We thank the reviewer for pointing us to these related papers, which we were not aware of
>     *   Safran et al. 2021. That paper provides an extension and additional results with respect to the cited paper Safran and Shamir 2018 by expanding the theoretical part of the analysis. In particular the study of the volume of the global minimum's basin of attraction. However, while those works identify that overparameterization helps, our paper identifies the exact geometric mechanism explaining _how_ spurious minima disappear. By mapping the dynamics into a low-dimensional space using mean-field summary statistics and employing the string method, we go beyond observing basin volumes to map the actual topological phase transition. We formally demonstrate how overparameterization creates flat directions that actively merge previously isolated minima into continuous, connected manifolds. Our framework isolates the "compensation mechanism" in the weight magnitudes required to unlock these flat paths. In summary, our analysis provides a more complete picture that goes beyond local geometry.
>     *   We agree that the work by Arjevani and Field should have been cited, and we include a detailed discussion in the revision. That said, there is an important technical distinction between their approach and ours. Both papers employ symmetry arguments to identify spurious minima and compute their loss values, but they do so in very different settings. Arjevani and Field focus on a local analysis in the high-dimensional weight space, using the Hessian to study curvature at specific critical points. In their framework, overparameterisation “annihilates” spurious minima by introducing a negative eigenvalue, effectively turning those points into saddles. In contrast, our work takes a global perspective by mapping the optimisation dynamics into a low-dimensional space of mean-field order parameters. This shift in viewpoint lets us move beyond local stability analysis. By following the macroscopic dynamics and applying the string method, we can directly probe the global structure of the loss landscape and identify a topological phase transition, namely spurious minima from being isolated become fully connected. From this perspective, overparameterisation does more than just destabilise isolated minima. It creates flat directions that connect them, forming continuous manifolds. Our order-parameter formulation also makes the mechanism behind this transition more transparent: the compensation mechanism shows how changes in weight magnitudes drive the geometry of the landscape. These kinds of global, mechanistic insights are not accessible from a purely local Hessian-based analysis.
> *   **"The studied model is too specific to be considered useful... restricts the significance and applicability of results."** While we acknowledge that the setting is limited, there are very few analytically tractable settings and results allowing us to study the impact of overparametrisation on non-convex neural network landscapes. This is one of them, and despite the several papers studying it (which is also a proof of its interest), there are still fundamental open questions. Our work identifies a regular geometric structure on this non-convex high-dimensional landscape, and shows the existence of a novel, topological phase transition. Furthermore, we do consider generalisations in the paper and appendix, such as a trainable second layer and different activations like Leaky ReLU and erf. We didn't test mismatched student-teacher activations and we have added a comment on this to our limitations section.

---

> > ### Author Rebuttal · Reviewer_emT6 · 2026-04-01
> >
> > Thank you for your response. After reading the reviews, I decide to maintain my score.

---

### Decision · Program_Chairs · 2026-04-30

**Decision:**

Accept (regular)

**Comment:**

There was extensive discussion on this paper during the review process.

Reviewer emT6’s main concerns were about some related existing work which have been largely addressed by the authors. Multiple reviewers (emT6, Xuot) commented about the assumptions used in the analysis such as the block-symmetric ansatz and orthonormal teachers. The authors have clarified that the former is an ansatz (and therefore not something that can be proved). The latter was simply an attempt at simplifying some of their analytical expressions. The mean-field calculations in the Appendix work for more general teachers also. The main point of content remaining seems to be that of exposition.

There was an extensive discussion with Reviewer 7QNk. The conclusion of this discussion was that the reviewer agrees that the paper has merit, both in terms of its analytical results and in terms of its novelty over existing work.

Reviewer WvpR also pointed out a few related papers which are related to this paper but address quite different aspects of the problem. The authors have provided a good rebuttal to some of the questions raised by Reviewer WvpR and the reviewer now agrees that this work provides valuable insights.

Reviewers 7QNk and Reviewer Xuot felt that the paper could benefit from being written differently. The authors have committed to doing this.

I am happy to recommend the paper for acceptance. I would suggest that the authors incorporate this feedback into the narrative.